# Targeting human Acyl-CoA:cholesterol acyltransferase as a dual viral and T cell metabolic checkpoint

Nathalie M. Schmidt [1], Peter A. C. Wing [2], Mariana O. Diniz[1], Laura J. Pallett [1], Leo Swadling [1], James M. Harris [2], Alice R. Burton[1], Anna Jeffery-Smith[1], Nekisa Zakeri [1], Oliver E. Amin [1], Stephanie Kucykowicz[1], Mirjam H. Heemskerk [3], Brian Davidson[4,5], Tim Meyer[5,6], Joe Grove [1], Hans J. Stauss [1], Ines Pineda-Torra [7], Clare Jolly [1], Elizabeth C. Jury [7], Jane A. McKeating [2] & Mala K. Maini [1]✉

Determining divergent metabolic requirements of T cells, and the viruses and tumours they fail to combat, could provide new therapeutic checkpoints. Inhibition of acyl-CoA:cholesterol acyltransferase (ACAT) has direct anti-carcinogenic activity. Here, we show that ACAT inhibition has antiviral activity against hepatitis B (HBV), as well as boosting protective anti-HBV and anti-hepatocellular carcinoma (HCC) T cells. ACAT inhibition reduces CD8[+] T cell neutral lipid droplets and promotes lipid microdomains, enhancing TCR signalling and TCR-independent bioenergetics. Dysfunctional HBV- and HCC-specific T cells are rescued by ACAT inhibitors directly ex vivo from human liver and tumour tissue respectively, including tissue-resident responses. ACAT inhibition enhances in vitro responsiveness of HBV-specific CD8[+] T cells to PD-1 blockade and increases the functional avidity of TCR-gene-modified T cells. Finally, ACAT regulates HBV particle genesis in vitro, with inhibitors reducing both virions and subviral particles. Thus, ACAT inhibition provides a paradigm of a metabolic checkpoint able to constrain tumours and viruses but rescue exhausted T cells, rendering it an attractive therapeutic target for the functional cure of HBV and HBV-related HCC.

[1] Division of Infection & Immunity, Institute of Immunity & Transplantation, University College London, London, UK. [2] Nuffield Department of Medicine, Oxford University, Oxford, UK. [3] Department of Hematology, Leiden University Medical Center, Leiden, The Netherlands. [4] Division of Surgery, University College London, London, UK. [5] Royal Free London NHS Foundation Trust, London, UK. [6] Cancer Institute, University College London, London, UK. [7] Division of Medicine, University College London, London, UK. ✉email: m.maini@ucl.ac.uk

Hepatitis B virus (HBV) causes an estimated 880,000 deaths a year from liver disease and hepatocellular carcinoma (HCC)[1]. Current treatment for chronic HBV infection (CHB) requires long-term antiviral suppression with nucleos(t)ide analogues (NUCs), which are unable to reduce transcription from the episomal and integrated forms of HBV DNA, resulting in ongoing expression of viral antigens and low rates of sustained off-treatment responses[2–4]. A pressing goal is to develop a combination of more potent antiviral and immunomodulatory approaches that can achieve functional cure of HBV, defined as loss of detectable circulating viral surface antigen (HBsAg)[2,3]. New treatments are needed to tackle the ongoing risk of HCC in patients on currently available antivirals[5]. Worldwide, HBV is the commonest cause of HCC, and HCC is the third most common cause of cancer-related deaths;[6] more effective treatments are therefore urgently needed[2–4].

T cells are capable of specifically recognising and eliminating virally infected or transformed hepatocytes; ongoing T cell immune surveillance is crucial to control transcription of residual episomal and integrated HBV DNA that cannot be eliminated by antiviral drugs, and to limit carcinogenesis and cancer progression[2,3,7,8]. Multiple checkpoints constrain the survival, expansion and function of virus- and tumour-specific T cells in the tolerogenic liver and tumour niche, providing potential targets for immunotherapies to rescue endogenous responses or to optimise adoptive cell therapies[3,9]. PD-1 blockade has been tested in phase III HCC trials and in a phase Ib study in HBV patients without HCC;[10,11] however, in both settings only a minority of patients had sustained responses, underscoring the need for additional approaches to increase therapeutic efficacy.

The central role of nutrient uptake and bioenergetic pathways in shaping successful immune responses has now been recognised and is leading to the exploration of the therapeutic potential of metabolic checkpoints[12]. However, many nutrients and metabolic pathways are required by both T cells and tumour cells, such that cutting off supplies to the tumour would also starve and constrain the T cells[13,14]. Here, we have explored the inhibition of cholesterol esterification by the enzyme acyl-CoA:cholesterol acyltransferase (ACAT1 and 2, also known as sterol O-acyltransferase, SOAT1 and 2) as an exception to this paradigm. Recent studies have shown that ACAT knockdown or inhibition can directly reduce the growth of several tumours including HBV-related HCC[15–17]. In contrast, ACAT1 knockdown boosted rather than constrained the anti-tumour potential of T cells in murine melanoma models[18]. We find that ACAT inhibition drives metabolic re-modelling, resulting in enhanced expansion and functionality of human CD8+ T cells directed against HBV and HCC, sampled directly from the site of disease. We show additional therapeutic benefits of this approach include its capacity to act in a complementary manner to PD-1 blockade, to enhance the functional avidity of TCR-gene-modified T cells and to exert antiviral effects against HBV.

## Results

### ACAT inhibition enhances peripheral and intrahepatic HBV-specific T cells.
We investigated the impact of inhibiting cholesterol esterification on the dysfunctional HBV-specific CD8+ T cell response that is characteristic of CHB. The addition of an ACAT inhibitor during the culture of peripheral blood mononuclear cells (PBMC) from patients with CHB increased the proportion of CD8+ T cells producing the antiviral cytokine interferon γ (IFNγ) in response to overlapping peptides (OLP) (Fig. 1a; gating strategy Supplementary Fig. 1a). A de novo IFNγ response could be induced by ACAT inhibition in a proportion of patients that lacked any detectable HBV-specific responses with

peptide stimulation alone (Supplementary Fig. 1b). For the cohort overall, there was a significant increase in both the fold change and absolute percentage of HBV-specific CD8+ T cells upon ACAT inhibition (Fig. 1b, Supplementary Fig. 1c). However, as reported for other immunotherapeutic strategies tested in vitro or in vivo[3,19], the response to ACAT inhibition was heterogenous; an enhanced IFNγ response was seen in patients with a spectrum of markers of disease activity (including some with viral load >2000 IU/ml; Supplementary Table 1). Of note, males were significantly more likely than females to respond to ACAT inhibition (Fig. 1c). Some patients showed an increase in other CD8+ T cell antiviral effector functions in response to ACAT inhibition (tumor necrosis factor (TNF) production increasing more consistently than degranulation marked by CD107a; Supplementary Fig. 1d, e). ACAT inhibition also increased IFNγ-producing HBV-specific CD4+ T cell responses in some donors but this effect was not significant for the whole cohort (Supplementary Fig. 1f).

The increase in functional HBV-specific CD8+ T cells was not merely due to the recovery of pre-existing responses, but also their expansion due to enhanced proliferation (as indicated by CFSE dilution of HBV-specific CD8+ T cells examined in a selected group of responders, Fig. 1d). The enhanced proliferation of CD8+ T cells was reproducible using two different ACAT inhibitors, Avasimibe (inhibiting ACAT1/2) and K-604 (ACAT1-specific) (Supplementary Fig. 1g). Importantly, ACAT inhibition did not induce non-specific cytokine production by unstimulated CD8+ T cells or perturb cell viability, nor did it further expand the highly functional CD8+ T cell responses to the well-controlled virus CMV in patients with CHB (Supplementary Fig. 1h–j).

As HBV replicates exclusively in hepatocytes, HBV-specific T cell responses need to function within the highly tolerogenic liver to control viral infection. By mining published single-cell (sc) RNA-Seq data[20], we first confirmed that ACAT1 (SOAT1) transcripts were detectable in equal percentages of intrahepatic and peripheral CD4+ and CD8+ T cells (whereas ACAT2 was barely detectable), supporting the potential for intrahepatic T cells to respond to ACAT inhibitors (Supplementary Fig. 1k, l). To test the potential of ACAT inhibition to act on immune responses at the site of infection, intrahepatic leucocytes (IHL) were isolated from HBV-infected liver tissue and stimulated overnight with OLP spanning the major HBV proteins core (HBc), surface (HBs) and polymerase (pol) (gating strategy Supplementary Fig. 1m). ACAT inhibition significantly enhanced antiviral IFNγ production by intrahepatic CD8+ T cells responding to peptides from all the major HBV antigens (calculated either as a proportion of CD8+ T cells or of total live lymphocytes, Fig. 1e, f; Supplementary Fig. 1n) and induced de novo HBV-specific IFNγ production in selected samples (Supplementary Fig. 1o). Increases in intrahepatic HBV-specific CD8+ T cells producing TNF or degranulating were less consistent (Supplementary Fig. 1p, q), suggesting ACAT inhibition is less likely to promote cytotoxic responses driving liver damage. However, there was a highly significant increase in IFNγ-producing CD4+ T cells directed against HBV within the liver (Supplementary Fig. 1r).

Taken together, ACAT inhibition tended to boost HBV-specific CD4+ and CD8+ T cells from the liver to a greater extent, and more consistently, than those from the blood (comparison of paired samples, Fig. 1g, h; Supplementary Fig. 1s) with only one donor failing to show an increase in intrahepatic responses to any HBV peptide pool tested. We postulated that the high local concentrations of cholesterol in the liver[21], a central hub for lipid metabolism, could contribute to this heightened sensitivity of local responses to ACAT inhibition. In support of this possibility,

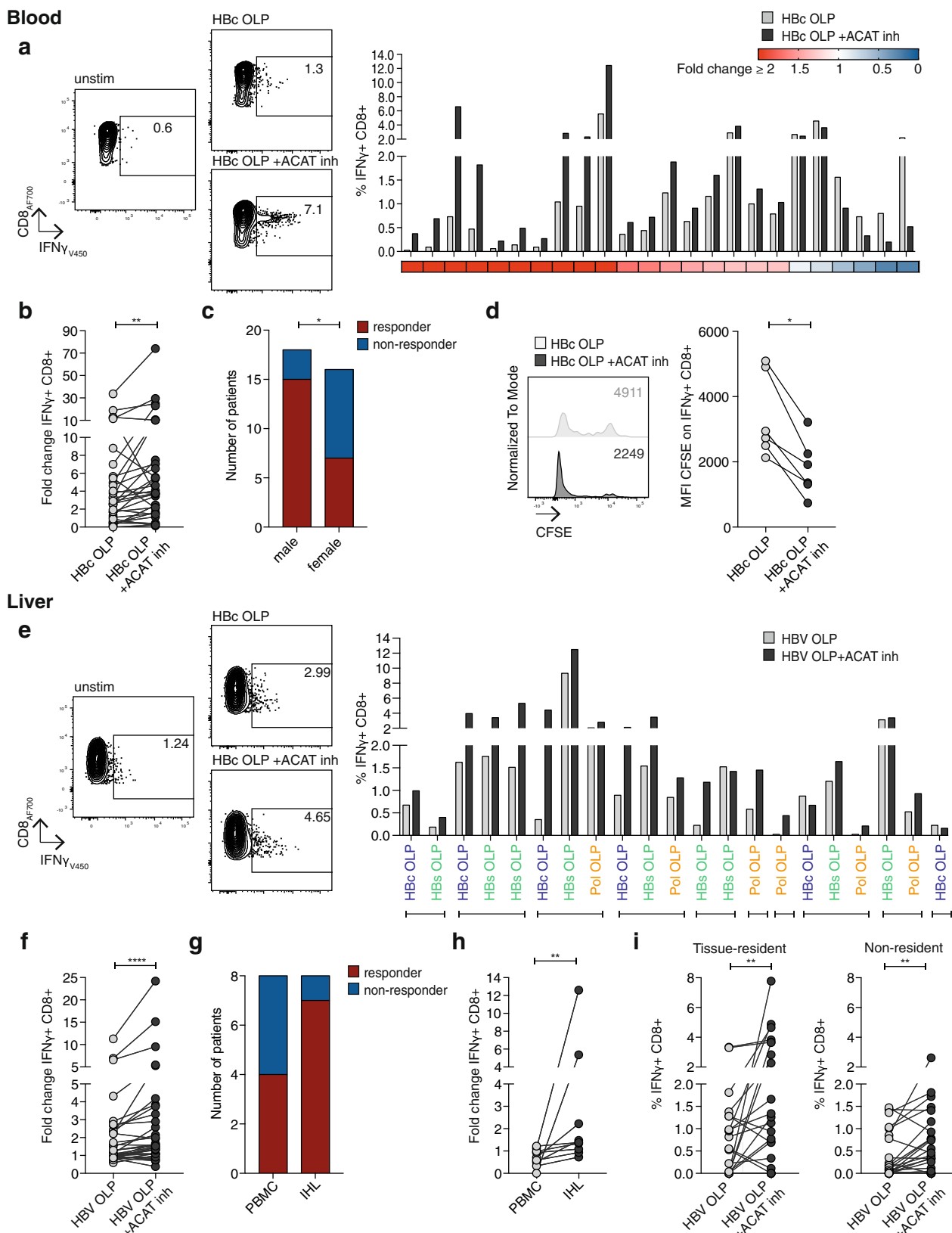

T cells showed a significantly enhanced proliferative response to ACAT inhibition after incubation in high cholesterol media (Supplementary Fig. 1t).

Within the pool of intrahepatic CD8$^+$ T cells, we recently reported a subset with the phenotype of tissue-resident memory

(CD69$^+$CD103$^+$) that are expanded in patients with efficient control of HBV, in line with their crucial role in frontline pathogen immune surveillance within non-lymphoid tissues[22]. ACAT inhibition did not alter the expression of these tissue retention markers (Supplementary Fig. 1u) but was able to

**Fig. 1 ACAT inhibition enhances peripheral and intrahepatic HBV-specific CD8+ T cells. a–d** PBMC from patients with CHB were stimulated with HBc overlapping peptides (OLP) ±ACAT inhibition (ACAT inh; Avasimibe or DMSO without Avasimibe for 8d). HBV-specific IFNγ production of CD8+ T cells was detected by flow cytometry. The IFNγ production in wells without peptide stimulation was subtracted to determine HBV-specific IFNγ production in summary data. **a** Representative flow cytometry plot and summary data for each patient with detectable pre-existing HBc OLP-specific CD8+ T cell response ($n = 24$). Heatmap indicates fold change of IFNγ production with peptide ±ACAT inhibition for each patient. **b** Fold change of IFNγ production after stimulation with HBc OLP ±ACAT inhibition normalised to unstimulated control. **c** Gender difference (cohort: male $n = 18$; female $n = 16$) in response to ACAT inhibition. Response defined as increased or de novo IFNγ production. **d** Assessment of proliferation of HBV-specific CD8+ T cells by CFSE dilution after gating on IFNγ+ responses; representative histogram and summary data ($n = 6$). **e–i** Intrahepatic leucocytes (IHL) from patients with CHB were stimulated with OLP spanning multiple HBV proteins (HBc, HBs, Pol) ±ACAT inhibition (K-604 or DMSO for 16 h). IFNγ production of CD8+ T cells was detected by flow cytometry with subtraction of background IFNγ production in unstimulated wells to determine HBV-specific IFNγ production in summary data. **e** Representative flow cytometry plot and summary data for each individual peptide pool ($n = 21$) and each patient ($n = 10$) with detectable pre-existing HBV OLP-specific CD8+ IHL responses. Brackets below the histogram indicate different OLP pools tested in IHL from the same patient. **f** Fold change of IFNγ production after stimulation with HBV OLP ±ACAT inhibition normalised to unstimulated control. **g, h** Paired PBMC and IHL stimulated with HBV OLP ±ACAT inhibition (K-604 or DMSO for 16 h). **g** Response to ACAT inhibition (patients $n = 8$). Response defined as increased or de novo IFNγ production. **h** Fold change of IFNγ production after stimulation with peptide +ACAT inhibition normalised to peptide +DMSO in patients with detectable pre-existing HBV-specific IFNγ production ($n = 9$ OLP pools and five donors). **i** IFNγ production ±ACAT inhibition by tissue-resident (CD103+CD69+) and non-resident (CD103−CD69−) CD8+IHL. $P$ values determined by Wilcoxon matched-pairs signed-rank test (**b, d, f, h, i**) and Fisher's exact test (**c, g**).

significantly enhance the function of the tissue-resident (CD69+ CD103+) as well as the non-resident (CD69−CD103−) fraction within intrahepatic CD8+ T cells (Fig. 1i), highlighting its capacity to boost responses capable of mediating long-lived local memory.

**ACAT inhibition induces metabolic re-wiring of CD8+ T cells.** We next explored the metabolic changes underpinning the rescue of CD8+ T cell function achieved by ACAT inhibition. Cholesterol is a major component of cell membranes and intracellular levels are tightly regulated by rates of uptake, export and synthesis[23,24]. Any excess cholesterol is esterified and stored in neutral lipid droplets in the cytoplasm[23,24]. However, an accumulation of lipid droplets can inhibit immune cell function[25–27]. As ACAT catalyses the esterification of cholesterol, its inhibition would be predicted to reduce the accumulation of neutral lipid droplets in T cells with potential beneficial effects on functionality. In line with this, the intensity of neutral lipid droplet (LipidTOX) staining in the cytoplasm of CD8+ T cells was reduced following ACAT inhibition (Fig. 2a). Neutral lipid droplets were already reduced within 1 h of ACAT inhibition, an effect that could be sustained by repeated treatment over 7 days (without any change in cell size, Supplementary Fig. 2a, b). Higher ex vivo levels of CD8+ T cell neutral lipid droplets correlated with a stronger response to ACAT inhibition (Supplementary Fig. 2c), underscoring lipid droplet reduction as one potential mechanism of action of ACAT inhibitors. T cells exposed to cholesterol in the presence of ACAT inhibition would be unable to esterify it and have been shown to instead mobilise more free cholesterol into the cell membrane[18]. In line with this, Filipin staining showed a subtle increase in unesterified membrane cholesterol upon ACAT inhibition when PMBC were incubated in high cholesterol media (Supplementary Fig. 2d).

Next, we examined whether ACAT inhibition could modulate the cholesterol- and glycosphingolipid-rich microdomains within the cell membrane that enable formation of the immunological synapse required for T cell receptor (TCR) activation following ligand engagement[28,29]. We visualised these lipid microdomains using fluorescent-labelled cholera toxin B subunit (CTB) that binds monosialotetrahexosylganglioside (GM1)[30,31]. We first confirmed that CTB staining co-localised with the TCR within the immune synapse formed between CMV-specific CD8+ T cells and an antigen-presenting cell (APC) line pulsed with their cognate peptide (Fig. 2b). We then applied flow cytometry to measure CTB within large numbers of CD8+ T cells at a single cell level to quantify the impact of ACAT inhibition on lipid

microdomains. Inhibiting ACAT significantly enhanced the proportion of CD8+ T cells with high levels of CTB staining (Fig. 2c) indicative of increased lipid microdomain formation. The magnitude of increase in CTB staining following ACAT inhibition associated with the degree of recovery of functional HBV-specific CD8+ T cells (Supplementary Fig. 2e).

To test whether ACAT inhibition affected T cell capacity for downstream TCR signalling, we carried out phosphoflow analysis of stimulated primary CD8+ T cells. The phosphorylation of several molecules downstream of TCR engagement (pERK, pAKT, Fig. 2d, e) was upregulated by ACAT inhibition, as was the phosphorylation of S6 ribosomal protein (Fig. 2f), demonstrating enhanced activity of the metabolic master-regulator mammalian target of rapamycin (mTOR).

To examine whether ACAT inhibition directly impacted the bioenergetics of T cells, we used PMA/ionomycin stimulation to assess this independently of the changes in mTOR activity observed following TCR signalling. The bioenergetics of purified human CD8+ T cells were analysed using the Seahorse extracellular flux analyser to simultaneously assess oxidative phosphorylation (OXPHOS, measured by oxygen consumption rate, OCR, with a mitochondrial stress test) and glycolysis (basal extracellular acidification rate, ECAR). ACAT inhibition resulted in significant increases in both basal OCR and ATP production (assessed using the complex V inhibitor oligomycin) and an increase in maximal respiration in five out of six donors (real-time OCR profile and summary data Fig. 2g). Although ACAT inhibition also significantly upregulated glycolysis, as shown by an increase in basal ECAR, the overall increase in OCR/ECAR ratio indicated a dominant effect on OXPHOS following a TCR-independent stimulus (Fig. 2h).

In summary, ACAT inhibition drove a redistribution of cholesterol so that less was esterified to be stored in neutral lipid droplets, and more was available to enhance lipid microdomains and TCR signalling. In addition, ACAT inhibition boosted TCR-independent bioenergetics, thus driving comprehensive metabolic re-programming to account for the observed enhancement in CD8+ T cell func6tionality.

**Complementary effects of ACAT inhibition and PD-1 blockade.** To further investigate how ACAT inhibition targets exhausted T cell responses such as those directed against HBV, we next probed the relationship between membrane lipids and immune checkpoints, such as PD-1. In line with their reduced function, PD-1+ CD8+ T cells showed markedly lower levels of CTB staining, suggesting an impaired capacity to assemble lipid

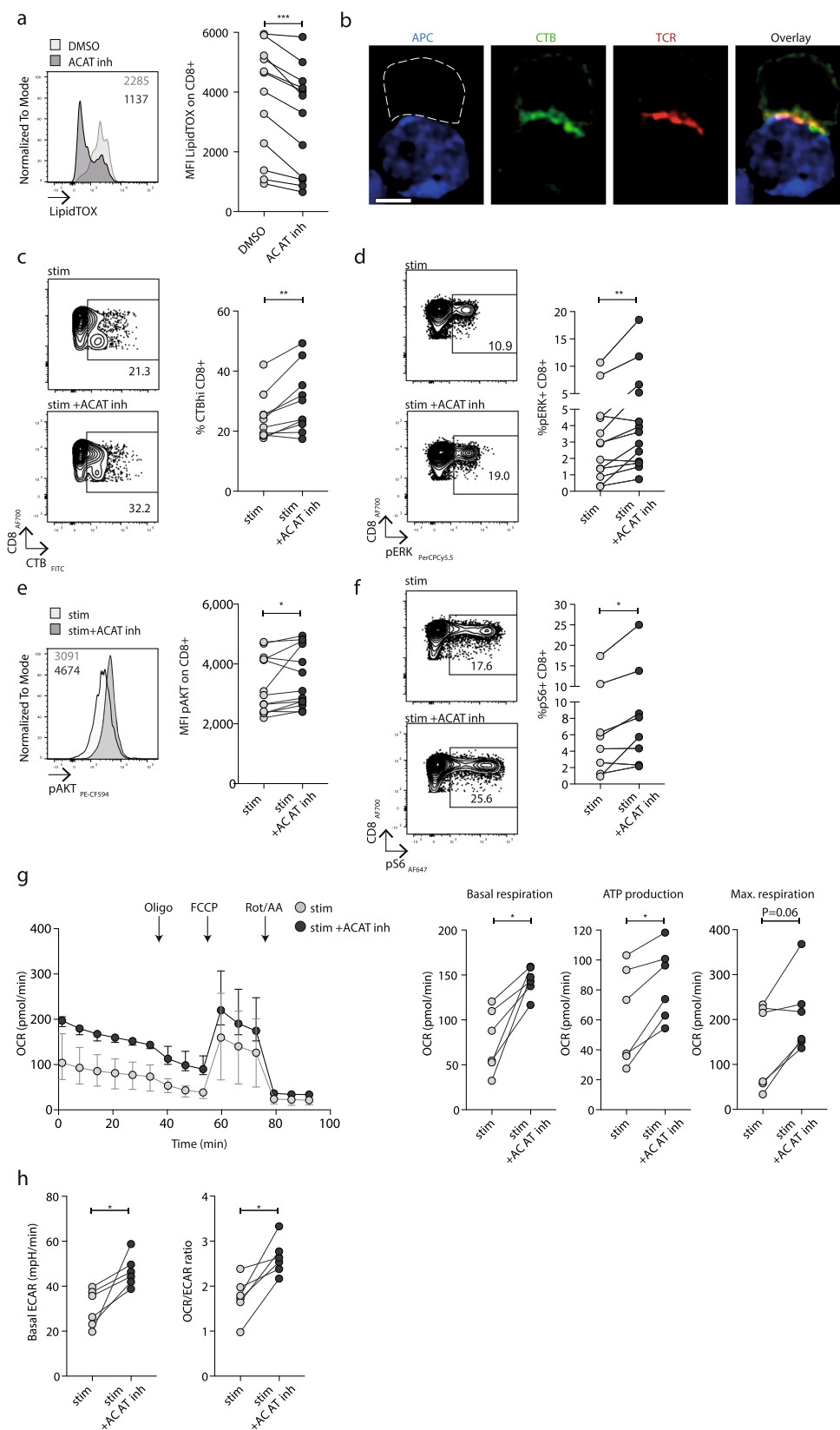

microdomains, compared to their PD-1[−] counterparts (mean percentage reduction 41%, Fig. 3a, mean MFI reduction 32%, Supplementary Fig. 3a). A similar reduction in CTB staining was seen using 2B4 as an alternative marker for exhausted CD8[+] T cells (mean percentage reduction 52%, Fig. 3a). However, checkpoints such as PD-1 can be upregulated in acutely activated

as well as exhausted human T cells; to dissect this further we co-stained CD8[+] T cells for PD-1 and the early activation marker HLA-DR. CD8[+] T cells with the exhaustion profile PD-1[+]HLA-DR[−] showed significantly reduced CTB staining compared with those with an acutely activated PD-1[+]HLA-DR[+] phenotype (Fig. 3b). In addition, CD8[+] T cells expressing the alternative

**Fig. 2 ACAT inhibition induces metabolic re-wiring of CD8+ T cells. a** Neutral lipid droplets (LipidTOX staining) in unstimulated PBMC ±ACAT inhibition (K-604 or equivalent concentration of DMSO for 1 h). Representative histogram and summary data ($n = 12$). **b** Confocal microscopy visualising the immunological synapse between a CMV-specific CD8+ T cell (white dashed outline, identified by CTB and TCR staining) and CMV peptide pulsed T2 cell (APC, blue) with staining of GM1-enriched microdomains (identified by CTB staining, green) and αβTCR (red). White scale bar: 5 mm. **c** CTB staining on PBMC from patients with CHB after stimulation with HBc OLP ±ACAT inhibition (Avasimibe or DMSO for 8d). Representative flow cytometry plot and summary data ($n = 10$). **d–f** Representative phosphoflow cytometry plots/histograms and summary data to detect phosphorylated TCR signalling molecules pERK ($n = 13$; **d**), pAkt ($n = 13$; **e**), pS6 ($n = 8$; **f**) ±ACAT inhibition (Avasimibe or DMSO for 8d) and stimulation with aCD3/aCD28. **g** Real-time oxygen consumption rate (OCR) of PMA/Ionomycin-stimulated purified CD8+ T cells ±ACAT inhibition (Avasimibe or DMSO for 16 h, $n = 6$). Compounds were added as indicated (oligomycin (Oligo), FCCP, rotenone+antimycin A (Rot/AA)) to determine basal respiration, ATP production and maximal respiration. OCR time course: median and interquartile range. **h** Basal extracellular acidification rate (ECAR) and basal OCR/ECAR ratio of PMA/Ionomycin-stimulated purified CD8+ T cells ±ACAT inhibition (Avasimibe or DMSO for 16 h, $n = 6$). P values determined by Wilcoxon matched-pairs signed-rank test.

acute activation marker CD38 had higher CTB staining than CD38$^-$ CD8+ T cells (Fig. 3b).

The primary target for PD-1-mediated inhibition is the co-stimulatory molecule CD28, ligation of which promotes lipid microdomain formation[28,32–34]. Consistent with this, CTB staining was markedly reduced in CD8+ T cells lacking CD28 (Fig. 3c, Supplementary Fig. 3b), with PD-1$^+$CD28$^-$ CD8+ T cells having even lower CTB staining than those that had only upregulated PD-1 (PD-1$^+$CD28$^+$) or downregulated CD28 (PD-1$^-$CD28$^-$) (Fig. 3c).

If the response to ACAT inhibition was partially dependent on enhancing lipid microdomains, we reasoned that donors with the lowest baseline levels would be more likely to respond. In support of this, CHB patients who showed a response to ACAT inhibition had significantly lower ex vivo levels of CTB and CD28 staining on their CD8+ T cells (Fig. 3d). As CD8+ T cells lacking CD28 were more likely to respond to ACAT inhibition, whereas those expressing CD28 are known to respond better to PD-1 blockade[35], we postulated combining these immunomodulatory approaches could be additive within individuals and/or non-redundant within cohorts.

We therefore compared the expansion of virus-specific IFNγ$^+$ CD8+ T cells from a cohort of patients with CHB after in vitro PD-1 blockade, ACAT inhibition or both. As shown for other immunotherapeutic approaches[3,19], responses were heterogenous, sometimes decreasing in vitro rather than increasing, although attrition of baseline responses tended to be more common with PD-1 blockade alone (Fig. 3e). Responses to these two approaches were non-redundant within the cohort, as some patients had better responses to PD-1 blockade and others to ACAT inhibition alone (white and black boxed patients in key below the histogram, Fig. 3e). ACAT inhibition and PD-1 blockade given together was additive in 11 out of 26 patients, in whom the strongest boosting of HBV-specific responses was seen with the combination (example flow cytometry plots and red boxed patients in summary below the histogram, Fig. 3e). The addition of ACAT inhibition to PD-1 blockade also significantly enhanced the fold increase in HBV-specific CD8+ T cells for the cohort compared with PD-1 blockade alone (Fig. 3f). Overall, therefore, PD-1 blockade and ACAT inhibition acted in a complementary manner to optimise efficacy within the cohort, with some patients responding optimally to one, the other or the combination of both.

To further examine the therapeutic relevance of this combination we took advantage of access to samples from a patient with HBV-related HCC who had received 2 weeks of anti-PD-1 therapy with Nivolumab. The expansion of cytolytic and non-cytolytic antigen-specific CD8+ T cells exposed to PD-1 blockade in vivo was strikingly enhanced following ACAT inhibition in vitro (Fig. 3g). Taken together, these data support the use of ACAT inhibition to enhance responsiveness to PD-1 blockade in the setting of chronic infection and HCC.

**ACAT inhibition boosts intratumoural and genetically engineered T cell responses.** To follow-up on the therapeutic utility of ACAT inhibition in HCC, it was important to evaluate whether this strategy could also rescue responses from tumour-infiltrating leucocytes (TIL), which are subject to multiple local inhibitory mechanisms. For example, excess cholesterol in the tumour microenvironment can drive the expression of checkpoint inhibitors through the induction of X-box binding protein 1 (XBP1)-dependent ER stress in T cells[36]; however, CD8+ T cells treated with ACAT inhibitors showed no consistent change in PD-1 expression (Supplementary Fig. 4a). Instead, ACAT knockdown or the inhibitor Avasimibe have been shown to promote anti-tumour T cell immunity in murine cancer models[18]. To investigate whether this effect would translate to human HCC, we first confirmed from available sc RNA-Seq data[20] that ACAT1 was transcribed in human HCC TIL, in fact in a slightly higher percentage than in peripheral T cells (Supplementary Fig. 4b, c). Next, to identify anti-tumour T cell responses in human HCC TIL, we focused on peptides from three major groups of tumour-associated antigens (TAA)[8]: oncofetal antigens (alpha-fetoprotein, AFP), cancer-testis-antigens (melanoma-associated antigen A1, MAGE-A1, and New York oesophageal squamous cell carcinoma-1, NY-ESO-1) and, in cases with HBV-associated HCC, viral antigens. TIL isolated from resected HCC and stimulated overnight in the presence of an ACAT inhibitor showed significantly enhanced IFNγ$^+$ TAA-specific responses (as a proportion of CD8+ T cells or of total live CD45$^+$ TIL, Fig. 4a, b; Supplementary Fig. 4d), as well as the induction of de novo CD8+ T cell responses in some patients that had no detectable peptide-specific IFNγ production when untreated (Supplementary Fig. 4e), with no changes in the frequency of global CD3$^+$ or CD8$^+$ TIL (Supplementary Fig. 4f). The benefit of ACAT inhibition was not limited to one group of TAA, with some boosted responses seen from all three groups of TAA (Fig. 4a), although responses were heterogeneous, with less consistent boosting using TNF and degranulation as readouts (Supplementary Fig. 4g, h). IFNγ$^+$ CD4$^+$ T cells directed against TAA were also expanded by ACAT inhibition, as were global CD8$^+$ TILs activated by anti-CD3/CD28 (Supplementary Fig. 4i, j). In addition, ACAT inhibition enhanced the function of TAA-specific CD8+ T cells in the surrounding liver tissue that might have the capacity to infiltrate and contribute to the anti-tumour response (Supplementary Fig. 4k). The expansion of TAA-specific CD8+ T cells with a tissue-resident phenotype within HCC (CD69$^+$CD103$^+$, Fig. 4c) supports their utility in providing long-lived local anti-tumour protection[37], whilst boosting of responses from TIL with the

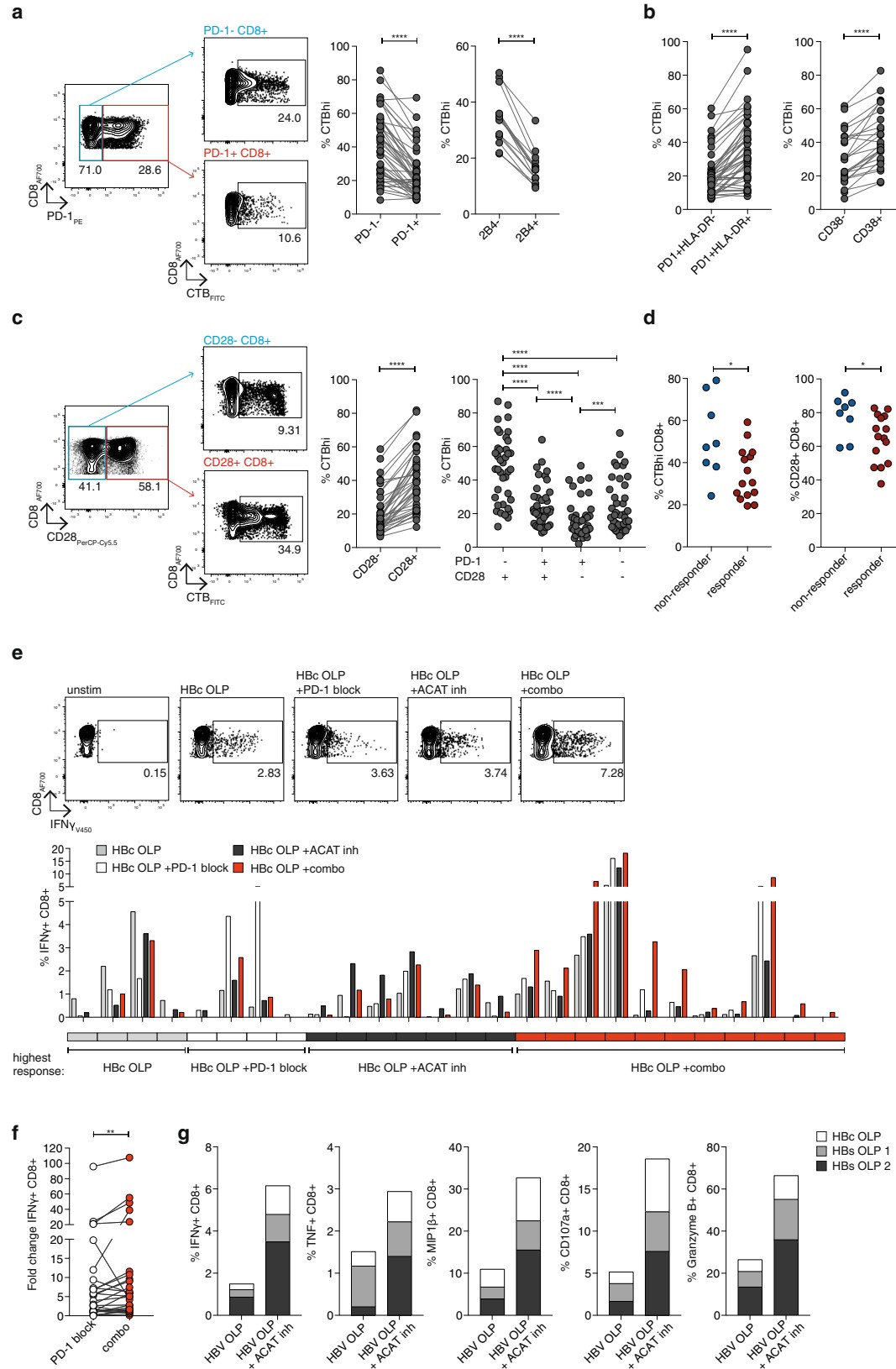

capacity to recirculate (CD69⁻CD103⁻, Fig. 4c) may be relevant for protection against the seeding of tumour metastases.

An alternative to therapeutic boosting of endogenous HCC immunity is to instead use adoptive cell therapy. For example, TCR-gene-modified T cells directed against HBV epitopes

expressed by HBV-related HCC have shown promising initial results in patients[38]. Next, we therefore tested whether ACAT inhibition could enhance the capacity of HBV-specific TCR-gene-modified CD8⁺ T cells to recognise and kill a hepatoma line (HepG2) presenting cognate peptide. Primary human T cells

**Fig. 3 Complementary effects of ACAT inhibition and PD-1 blockade. a–c** Ex vivo CTB staining of PBMC from patients with CHB to identify GM1-enriched microdomains. **a** Representative flow cytometry plot and summary data of %CTBhi of PD-1$^+$ ($n = 46$) and 2B4$^±$ ($n = 15$) CD8$^+$ T cells. **b** %CTBhi of exhausted (PD-1$^+$ HLA-DR$^-$) vs activated (PD-1$^+$ HLA-DR$^+$) CD8$^+$ T cells ($n = 46$) and of activated (CD38$^+$) vs non-activated (CD38$^-$) CD8$^+$ T cells ($n = 25$). **c** Representative flow cytometry plot and summary data of %CTBhi of CD28$^±$ CD8$^+$ T cells and CD28$^±$ PD-1$^±$ subsets ($n = 39$). **d** Ex vivo %CTBhi CD8$^+$ T cells and %CD28$^+$ CD8$^+$ T cells according to response to ACAT inhibition ($n = 23$). Response defined as increased or de novo HBV-specific IFNγ production. **e, f** PBMC from patients with CHB stimulated with HBc OLP in the presence of DMSO (grey), ACAT inhibition (Avasimibe, black), PD-L1/PD-L2 blockade +DMSO (PD-1 block, white) or a combination of ACAT inhibition and PD-L1/PD-L2 blockade (combo, red) for 8d. **e** Representative flow cytometry plots and summary data of patients showing IFNγ production in any of the four conditions ($n = 26$). Boxes below the histogram indicate treatment strategy resulting in the highest IFNγ production in the respective patients (grey: DMSO; white: PD-1 block; black: ACAT inhibition; red: combo). **f** Fold change of IFNγ with PD-1 block or combo normalised to control without peptide stimulation. **g** Effector function of PBMC from a patient with HBV-related HCC 2 weeks after start of in vivo anti-PD-1 immunotherapy. PBMC stimulated with HBV OLP ±ACAT inhibition (Avasimibe or DMSO for 8d). $P$ values determined by Wilcoxon matched-pairs signed-rank test (**a–c** (left), **d–f**) or Friedman test with Dunn's multiple comparisons test (**c** (right)).

transduced with a TCR specific for an HLA-A2-restricted HBV epitope (kindly provided by A. Bertoletti, Singapore)[39] had enhanced capacity to produce the anti-tumour cytokine IFNγ in response to limiting concentrations of their cognate peptide pulsed on the hepatoma cell line, following ACAT inhibition (increased percentage and MFI IFNγ$^+$ by intracellular cytokine staining, Fig. 4d; Supplementary Fig. 4l, increased supernatant IFNγ by Luminex, Supplementary Fig. 4m). ACAT inhibition of TCR-gene-modified T cells similarly enhanced the production of other immune mediators with potential anti-tumour effects (TNF, MIP1β by intracellular cytokine staining, Supplementary Fig. 4n, o; TNF, MIP1β, Flt-3L, CD40L by Luminex, Supplementary Fig. 4m). ACAT inhibition also enhanced specific lysis of the hepatoma target cell line, a crucial feature for efficient antitumor function (Fig. 4e). Preliminary data suggested ACAT inhibition was also able to increase the functional avidity of gene-modified T cells specific for the minor histocompatibility antigen HA-1, which is known to be expressed at the cell surface with low efficiency[40]. ACAT inhibition reduced the minimum peptide concentration required to trigger antigen-specific T cell effector function, resulting in a comparably enhanced functional avidity to that achieved by the recent framework engineering of this TCR by LYR modification[41] (Supplementary Fig. 4p).

Taken together, ACAT inhibition increased the anti-tumour potential of endogenous TAA-specific CD8$^+$ T cells within human HCC examined directly ex vivo, as well as increasing the functional avidity of TCR-gene-modified CD8$^+$ T cells.

**ACAT regulates de novo HBV particle genesis.** A final aspect we considered for the therapeutic application of ACAT inhibitors in HBV and HBV-related HCC was whether they could exert any direct effect on HBV replication independent of the immune response. ACAT is highly expressed in hepatocytes[42] and cholesteryl ester is one of the components of HBsAg[43]. We, therefore, hypothesised that inhibiting cholesterol esterification in infected hepatocytes may interfere with the formation of new HBsAg-containing virions and subviral particles.

We examined the impact of inhibiting ACAT on HBV replication in human HepG2 hepatocyte derived-cells engineered to express the viral entry receptor, sodium-taurocholate co-transporter polypeptide (NTCP)[44]. Building on our finding that ACAT regulates the cholesterol composition of cell membranes, we initially assessed the impact of inhibiting ACAT on HBV uptake into HepG2-NTCP cells using recently reported methodology[45]. Treating HepG2-NTCP cells with ACAT inhibition induced a modest twofold increase in internalised virus (Fig. 5a). To confirm that HBV uptake was NTCP-dependent, we showed that a synthetic peptide mimicking the preS1-binding site for NTCP, Myrcludex-B (MyrB)[46], inhibited HBV internalisation. However, culturing the infected cells in the continued presence of

the ACAT inhibitor, showed no significant change in cccDNA levels or HBeAg (Fig. 5b). ACAT inhibition had no cytotoxic effects on HepG2-NTCP cells (Supplementary Fig. 5a). However, we noted a significant dose-dependent reduction in extracellular HBV DNA (Fig. 5b), where the reduction in de novo particle genesis was comparable to that seen with a standard-of-care NUC, Entecavir. Importantly, ACAT inhibition also reduced extracellular HBsAg, an effect not seen with Entecavir (Fig. 5b).

We then examined the impact of initiating treatment following, rather than during, the establishment of HBV infection in HepG2-NTCP cells. Of note, ACAT inhibition significantly reduced both secreted HBV DNA and HBsAg in this setting (Fig. 5c). We confirmed the antiviral effect of ACAT inhibition on HepG2 cells transduced with an adenoviral vector encoding the HBV genome[47]. This system allows efficient transfer of the HBV genome into cells resulting in high viral transcription rates and genesis of infectious HBV particles. Inhibiting ACAT in Ad-HBV transduced HepG2-NTCP cells resulted in a significant dose-dependent reduction in extracellular HBV DNA levels, confirming our earlier observation (Supplementary Fig. 5b). Together, these data show a clear antiviral effect of ACAT inhibition, reducing both extracellular HBsAg and HBV particle production, a dual effect that cannot be achieved by the antivirals currently used for the treatment of CHB. Further work is needed to investigate the antiviral effect of ACAT inhibitors in the HBV-infected liver in vivo. Analysing published microarray data for ACAT1 + 2 (SOAT1 + 2), we noted an increase in ACAT1 transcripts in HBV-infected compared with uninfected livers[48] (Supplementary Fig. 5c), particularly in cases with liver inflammation within CHB cohorts (ALT > 40 IU/l[48,49], Supplementary Fig. 5d, or "immune clearance" stage[50], Supplementary Fig. 5e); this raises the possibility that ACAT inhibitors will have differential effects in the setting of HBV-related inflammation.

In summary, these data highlight the potential of ACAT inhibitors to modulate HBV antigen burden beyond existing antiviral agents, an important effect complementary to their capacity to enhance antiviral and antitumour T cell immunity.

## Discussion

Overlap in the metabolic requirements of T cells with those of tumours or viruses can drive competition between them, contributing to immune paralysis. Identifying metabolic check-points that could be targeted therapeutically to block carcino-genesis without constraining immunity is a major therapeutic challenge[13,14]. ACAT inhibitors have already been shown to exert tumour-intrinsic effects to restrict the growth of several cancers in preclinical models, including early-stage HBV-related HCC[15-17]. We show for the first time that ACAT inhibition boosts human antigen-specific responses, rescuing CD8$^+$ T cells directed against HBV or tumour antigens after just overnight

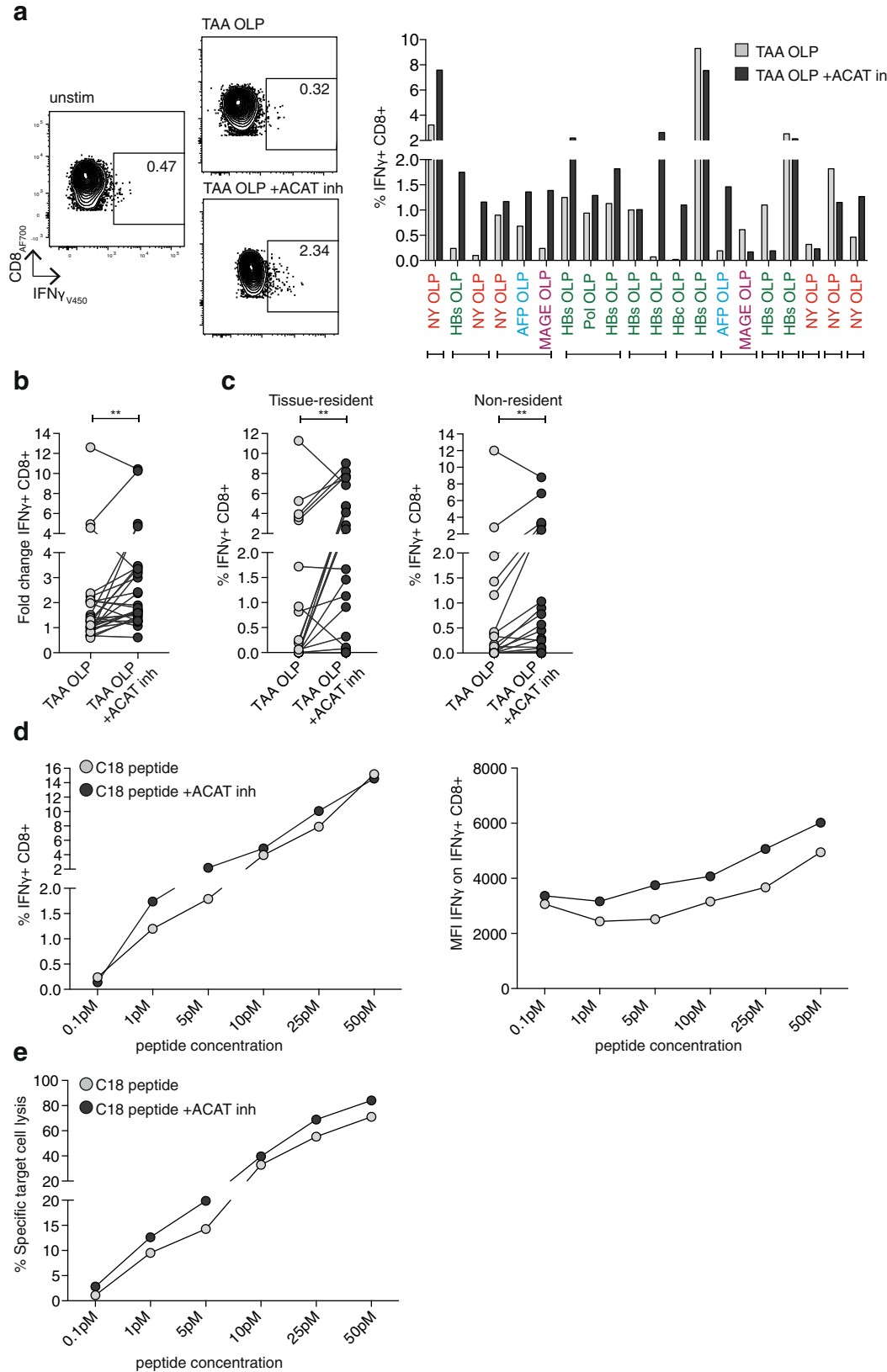

treatment of freshly isolated tissue samples. Our finding that ACAT inhibition can additionally act directly to reduce HBV particle genesis and secreted HBsAg release gives it the unique potential to exert multiple complementary modes of action in the therapy of HBV and HBV-related HCC. Our data support the

enzyme ACAT being an exception to most metabolic checkpoints, with its inhibition constraining tumours and viruses whilst conversely boosting T cells.

The liver, and many tumours, provide microenvironments that are rich in cholesterol[21,23,36], which is required by cells as a major

**Fig. 4 ACAT inhibition boosts intratumoural and genetically engineered T cell responses. a–c** Tumour-infiltrating leucocytes (TIL) from patients with HCC were stimulated with tumour-associated antigen (TAA; NY-ESO-1, MAGE-A1, AFP, HBc, HBs, Pol) OLP ±ACAT inhibition (K-604 or equivalent concentration of DMSO for 16 h) and IFNγ production of CD8$^+$ T cells was detected by flow cytometry. The IFNγ production in wells without peptide stimulation was subtracted to determine TAA-specific IFNγ production in summary data. **a** Representative flow cytometry plot and summary data for each individual peptide pool ($n = 20$) in patients ($n = 12$) with detectable pre-existing TAA-specific CD8$^+$ TIL responses. Brackets below the histogram indicate different OLP tested in TIL from the same patient. **b** Fold change of IFNγ production after stimulation with TAA OLP ±ACAT inhibition normalised to control without peptide stimulation. **c** IFNγ production of tissue-resident (CD103$^+$CD69$^+$) and non-resident (CD103$^-$CD69$^-$) CD8$^+$ TIL ±ACAT inhibition. **d, e** IFNγ production (**d**) and specific target cell lysis (**e**) by HBcAg18-27 -TCR-gene-modified CD8$^+$ T cells cocultured with HepG2-NTCP cells pulsed with increasing doses of C18 peptide (0.1 pM–50 pM) ±ACAT inhibition (Avasimibe or DMSO for 16 h). Representative data of four independent experiments. $P$ values determined by Wilcoxon matched-pairs signed-rank test.

component of lipid bilayers for membrane fluidity[23]. ACAT catalyses the esterification of excess intracellular free cholesterol for storage in neutral lipid droplets, that emerging data reveal can have opposing effects in tumours and immune cells. Whereas cholesteryl ester-containing lipid droplets can drive cancer cell proliferation and invasiveness[17], in immune cells excess neutral lipid droplets associate with impaired function[25–27]. We found that ACAT inhibition was capable of reducing neutral lipid droplets in human T cells by blocking the esterification of cholesterol and diverting it to the cell membrane to enhance lipid microdomain formation and TCR signalling, resulting in enhanced functionality. Cholesterol has also been shown to bind to the TCRβ-chain to stabilise TCR nanoclusters, enhancing T cell avidity upon ligand binding[51] whilst reducing inappropriate TCR triggering in the resting state[52]. These roles for membrane cholesterol are consistent with our observation that ACAT inhibition promoted the functional avidity of TCR-gene-modified T cells. Beyond these effects, we discovered that ACAT inhibition can also trigger TCR-independent boosting of glycolysis and OXPHOS, thereby optimising CD8$^+$ T cell bioenergetics to support proliferation and effector function.

Unlike ACAT inhibition, current standard-of-care NUCs do not have the capacity to directly modulate T cells, nor to reduce the release of subviral particles containing HBsAg[2,4]. A therapeutic reduction in circulating concentrations of HBsAg could limit it acting as a decoy for anti-HBs antibodies, as well as reducing the antigen-driven dysfunction of HBsAg-specific B cells[4]. Cholesteryl ester is an essential component of HBsAg[43], suggesting a potential mechanism underlying our observation that ACAT inhibition limits the genesis of both infectious virions and subviral particles in vitro. Recent reports that apolipoproteins associate with HBV and positively regulate particle infectivity raise the possibility of ACAT inhibition acting in an analogous manner to its activity reducing hepatitis C lipoviral particles[53–55].

Our findings raise many novel areas for future studies. It will be of interest to examine if the modest increase in HBV entry seen with ACAT inhibition is the result of an increase in lipid microdomains in hepatocytes, as we observed in T cells, as these lipid rafts are known to be important for the entry of many viruses[56]. This finding points to a likely synergy in the mechanism of action of ACAT inhibition with HBV entry inhibitors like Myrcludex B, which could be tested in preclinical models that allow spreading infection. In order to fully understand the effect of ACAT inhibition in vivo, further investigation of all relevant cellular targets of ACAT inhibitors in the HBV-infected liver is required, in particular, whether they can also modulate APC function. Further studies are also needed to define the triggers for the novel T cell bioenergetic changes we observed, which could reflect secondary effects of ACAT inhibition on sterol metabolic checkpoints like Liver X receptor (LXR) and SREBP[57–60]. In addition, our finding that human PD-1$^{hi}$CD28$^{lo}$ CD8$^+$ T cells have reduced lipid microdomain staining, and that this serves as a predictor of response to ACAT inhibition, raises the possibility

that ACAT will be a tractable metabolic checkpoint in other disease settings characterised by exhausted T cells, including other chronic viral infections and tumours.

Current trials of combinations of antiviral drugs and immunomodulatory agents for HBV have not been able to consistently induce functional cure (sustained loss of serum HBsAg), nor prevent carcinogenesis[2–5]. ACAT inhibitors, with the potential for a combination of immunomodulatory, antiviral, and anticarcinogenic effects, therefore provide an attractive addition to available agents for HBV and HBV-related HCC. The ACAT inhibitor Avasimibe was reported to be well-tolerated in phase III trials for atherosclerosis[61] and is concentrated in the liver following oral delivery[62]. Both HBV- and tumour-specific T cells are constrained by multiple pathways, such that the response to a single immunotherapeutic approach tends to be limited to subgroups. The immune-boosting capacity of ACAT inhibition could be used to improve the response to other immunotherapies being tested in patients with HBV and HCC; in particular, our data provide a mechanistic basis for, and ex vivo demonstration of, its capacity to boost the number of patients responding to PD-1 blockade. ACAT inhibition has the advantage of preferentially rescuing T cells from high cholesterol microenvironments like the liver and tumours. Our in vitro data suggest it may be possible to predict clinical response to ACAT inhibitors based on male gender and low T cell expression of CD28 or CTB. The ability of ACAT inhibitors to enhance CD8$^+$ T cell avidity is of particular relevance to HBV and tumours, as in both settings residual T cells are of low affinity and/or antigen presentation is inefficient[63–66]. Our results suggest that ACAT inhibition could also be applied to optimise TCR-gene-modified T cells and other adoptive cell therapies being developed for HBV and tumours including HCC, as in a recent study using click chemistry to deliver Avasimibe to CAR-T cells[67]. The finding that increased ACAT expression is an early feature of rapidly progressing HBV-related HCC[15] makes this a particularly compelling setting for initial clinical testing of its combined immunomodulatory, anticarcinogenic and antiviral efficacy. Our results exemplify the importance of studying the differential effects of the metabolic landscape on immune responses, tumours and viruses.

## Methods

**Ethical approval**. This study was approved by the local ethics boards (Wales Research Ethics Committee 4 and UCL Biobank Ethical Review Committee: Research Ethics Committee reference number 16/WA/0289; Brighton and Sussex: Research Ethics Committee reference number 11/LO/0421) and complies with the declaration of Helsinki. All storage of samples obtained complied with the Human Tissue Act 2004.

**Samples**. Peripheral blood samples were taken from healthy control individuals, patients with CHB and patients with HCC. Resected tissue from HBV-infected livers, liver tumours and unaffected surrounding liver, and paired blood samples were obtained through the Tissue Access for Patient Benefit (TapB) scheme at The Royal Free Hospital, London. During the analysis, the investigators were blinded to all clinical patient data. All study participants gave written informed consent prior to participation.

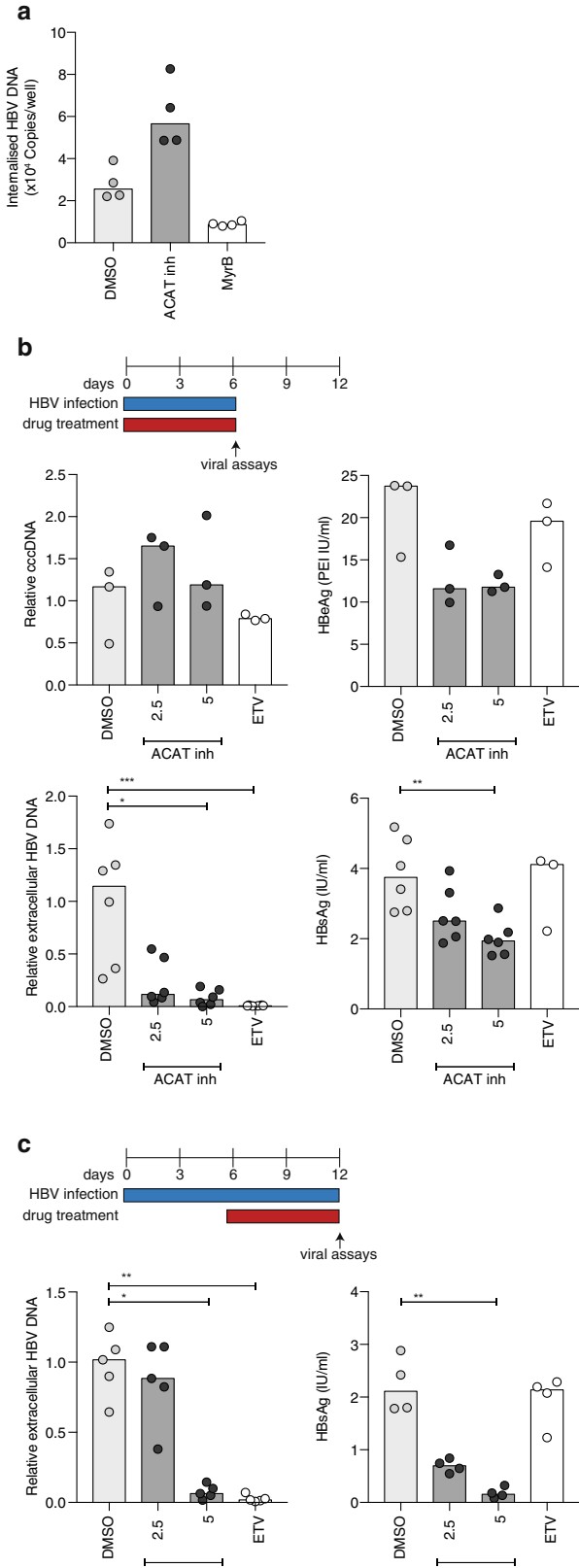

**Fig. 5 ACAT regulates de novo HBV particle genesis. a** HepG2-NTCP were infected with HBV in the presence of ACAT inhibition (Avasimibe), Myrcludex-B (MyrB) or DMSO. Viral uptake in 6 h was quantified by qPCR of intracellular HBV DNA. **b** HepG2-NTCP cells were infected with HBV (MOI 200) in the presence of ACAT inhibition (2.5 mg/ml and 5 mg/ml Avasimibe), Entecavir (ETV) or DMSO in two independent experiments with a total of six replicates. cccDNA (relative to untreated mean), HBeAg, extracellular HBV DNA (relative to untreated mean) and HBsAg were measured after 6d by either qPCR or ELISA. **c** HBV-infected HepG2-NTCP cells (6d post infection) were treated with ACAT inhibition (Avasimibe), ETV or DMSO in four independent replicates. Extracellular HBV DNA (relative to untreated mean) extracellular HBsAg were measured after 6d. *P* values determined by Kruskal–Wallis test with Dunn's multiple comparisons test. Bars indicate median values.

GentleMACS (Miltenyi Biotech), filtration through a 70 µm cell strainer (BD Bioscience), removal of parenchymal cells on a 30% Percoll gradient (GE Healthcare) and lymphocyte isolation via density centrifugation as described above. IHL and TIL were used immediately after isolation.

**Overnight and short-term cell culture**. To examine HBV- and HCC-specific CD8+ T cell responses, IHL isolated from HBV-infected livers were stimulated with HBV-derived OLP (15mers overlapping by 12 amino acids, pan-genotypic HBcAg (94 peptides), pan-genotypic HBsAg (divided into two pools with 84 and 82 peptides), gifted by Gilead Sciences, and genotype-D derived polymerase (Pol), 15mers overlapping by 10 amino acids (165 peptides), thinkpeptides). IHL from tumour-surrounding tissue and TIL were stimulated with TAA-derived OLP (15mers overlapping by 12 amino acids; NY-ESO-1 (43 peptides), AFP (150 peptides) and MAGE-A1 (75 peptides), JPT Peptide Technologies) and TIL from HBV-associated HCC were stimulated with HBV and TAA OLP. IHL, TIL and paired PBMC (where available) were stimulated with 1 µg/ml OLP for 16 h at 37°C in the presence of 1 µg/ml Brefeldin A (Sigma-Aldrich) and Monensin (BD Biosciences, 1:1500 dilution) in cRPMI (RPMI 1640 (Thermo Fisher Scientific)+2% HEPES buffer solution, 0.5% sodium pyruvate, 0.1% 2-mercaptoethanol, MEM 1% non-essential and 2% essential amino acids (Gibco) and 100U/ml penicillin/ streptomycin (Life Technologies)) and 10% heat-inactivated FBS (Sigma-Aldrich), followed by antibody staining and flow cytometric analysis. Where indicated, TIL were stimulated with 0.5 µg/ml plate-bound anti-CD3 (Invitrogen) and 5 µg/ml soluble anti-CD28 (Invitrogen) for 16 h. Cells were treated with 0.1 µM of the ACAT inhibitor K-604 (Sigma-Aldrich) or equivalent concentration of DMSO. Where sufficient cell numbers were available, experiments were performed in at least duplicates and combined prior to antibody staining. Due to limited cell numbers, in individual cases, not all peptide pools could be tested for all patient samples. The exact peptide pools tested for each patient are indicated in the respective figures.

To examine HBV-specific CD8+ T cell responses in the blood, PBMC were stimulated with 1 µg/ml HBc OLP in cRPMI+ 10% FBS+ 20 U/ml recombinant human IL-2 (PeproTech). PBMC were expanded at 37°C for 8d ± 0.5 µM of the ACAT inhibitor Avasimibe (Selleckchem) or equivalent concentration of DMSO replenished every 2d ± 2.5 µg/ml anti-PD-L1 and 2.5 µg/ml anti-PD-L2 (eBioscience) on d0 where indicated. On d7, PBMC were restimulated with 1 µg/ml peptide in the presence of Brefeldin A for 16 h at 37°C, followed by antibody staining and flow cytometric analysis. To examine CMV-specific CD8+ T cell responses, PBMC from HLA-A2+ donors were stimulated with 1 µg/ml of the HLA-A2-restricted immunodominant peptide CMVpp65$_{495-503}$ (NLV, amino-acid sequence: NLVPMVATV) at the start of the culture and for restimulation on d7. All experiments were performed in duplicates and combined prior to restimulation. The cytokine production in wells without peptide stimulation was subtracted to determine peptide-specific cytokine production in all summary data. A response to ACAT inhibition was defined as increased or de novo IFNγ production compared to DMSO. For the evaluation of cell proliferation, PBMCs were labelled with 1 µM CFDA prior to the start of culture. Where indicated, culture media was supplemented with cholesterol (SyntheChol 1:500, Sigma-Aldrich).

**TCR-gene-modified T cells**. CD8+ T cells were genetically engineered to express a TCR specific for the immunodominant HBc-derived epitope HBc$_{18-27}$ (C18; amino-acid sequence: FLPSDFFPSV)[39] or a TCR specific for an HA-1-derived epitope (amino-acid sequence: VLHDDLLEA; wildtype TCR and codon-optimised LYR-TCR as previously described)[41].

Phoenix amphotropic packaging cells (ATCC CRL-3213) were transiently co-transfected using FuGENE (Promega) with plasmids encoding the C18-specific TCR and an amphotrophic envelope. Retroviral supernatants were collected for the transduction of HLA-A2-negative healthy donor PBMC stimulated with 200 U/ml IL-2 and 50 ng/ml plate-bound anti-CD3. Activated lymphocytes were plated on

**PBMC, IHL and TIL isolation**. PBMC were isolated from heparinized blood by density centrifugation with Ficoll-Hypaque Plus (GE Healthcare) or Pancoll (Pan Biotech). PBMC were used fresh or were cryopreserved in 10% dimethyl sulfoxide (DMSO) (Sigma-Aldrich) prior to further use.

To isolate IHL and TIL, liver/tumour tissue was cut into small pieces and incubated for 30 min at 37°C in 0.01% collagenase IV (Invitrogen) and 0.001% DNAse I (Sigma-Aldrich), followed by further mechanical disruption via

retronectin (Takara) coated wells and mixed with retroviral supernatants. Cells were monitored for the expression of the endogenous recombinant TCR and after expansion in 100 U/ml IL-2, 10 ng/ml IL-7 and 10 ng/ml IL-15 (Peprotech), C18-TCR or HA-1-TCR-expressing CD8+ T cells were sorted with a FACSAria II (BD Bioscience).

To evaluate the function of the TCR engineered CD8+ T cells, HepG2 cells (for C18-TCR; ATCC HB-8065) or T2 cells (for HA-1 TCR; ATCC CRL-1992) were pulsed with the indicated doses of peptide and treated with 1 μM Avasimibe in cRPMI+10% FBS+100 U/ml IL-2, 10 ng/ml IL-7 and 10 ng/ml IL-15 at 37℃. To determine cytokine production by antibody staining and flow cytometric analysis, Brefeldin A and Monensin were added for 16 h. The cytokine production in wells without peptide stimulation was subtracted to determine peptide-specific cytokine production in summary data. ToxiLight Non-Destructive Cytotoxicity BioAssay (Lonza) measuring relative light units (RLU) was performed as described by the manufacturer to evaluate specific target cell lysis, calculated as (RLU(effector + target)−RLU(effector)−RLU(target))/(RLU(100%lysis)−RLU(effector)−RLU (target))*100.

The concentrations of immune mediators in the supernatant were measured by MAGPIX (Luminex, USA) using the human XL cytokine discovery 45-plex kit (Biotechne) according to the manufacturer's instructions. The assay was performed in three independent experiments and soluble mediators with consistent changes in all three experiments were included in the summary data. The immune mediator concentration in wells without peptide stimulation was subtracted to determine peptide-specific immune mediator concentration in summary data.

**Surface and intracellular staining.** For flow cytometry, cells were stained with saturating concentrations of surface antibodies and a fixable viability dye diluted in 1:1 PBS (Invitrogen): Brilliant Violet Buffer (BD Bioscience). Following surface staining, cells were either fixed with Cytofix (BD Bioscience) or fixed and permeabilized with cytofix/cytoperm (BD Bioscience) where further intracellular staining was performed. Intracellular antibodies in saturating concentrations were diluted in a 0.1% saponin-based buffer (Sigma-Aldrich).

For the detection of GM1-enriched microdomains, cells were surface stained as described above and then stained with 25 μg/ml cholera toxin B-fluorescein isothiocyanate (CTB-FITC) at 4℃ followed by fixation as above. For the detection of neutral lipid droplets, cells were surface stained and fixed as described above, followed by staining with HSC LipidTOX Neutral Lipid Stain according to the manufacturer's instructions. For the measurement of membrane cholesterol, cells were surface stained and fixed as described above, followed by a staining with 50 μg/ml Filipin complex (Sigma-Aldrich) for 2 h at room temperature.

All samples were acquired on a BD Bioscience Fortessa-X20 and analysed using FlowJo v.10 (BD Bioscience). Full details on monoclonal antibodies and other fluorescent agents can be found in Supplementary Table 2.

**Phosphoflow.** After pre-treatment with ACAT inhibitors for 7d as described above, PBMC were surface stained followed by a stimulation with 0.5 μg/ml plate-bound anti-CD3 (Invitrogen) and 5 μg/ml soluble anti-CD28 (invitrogen) for 30 min at 37℃. Cells were fixed and permeabilized with PhosphoFix/Perm Buffer (BD Bioscience) for 30 min at −20℃ followed by intranuclear staining with saturated concentrations of phospho-antibodies in PBS.

**Confocal microscopy.** PBMC were stimulated with 1 μg/ml of the HLA-A2-restricted immunodominant peptide CMVpp65$_{495-503}$ (NLV) and expanded in a short-term culture for 8d as described above. On d8 CD8+ T cells were isolated using a negative CD8+ T cell isolation kit (Miltenyi Biotech) according to the manufacturer's guidelines. T2 cells were pulsed with 1 μg/ml NLV peptide to function as APC.

APC were labelled with Cell Tracker Deep Red (Invitrogen) according to manufacturer's instructions. CD8+ T cells and APC were resuspended in cRPMI with 1% FBS in a 1:1 ratio and incubated on poly-L-lysine coated coverslips for 30 min at 37℃. Cells were then fixed at 4℃ in PBS with 4% formaldehyde and 1% bovine serum albumin (BSA) followed by a staining with 10 μg/ml CTB-FITC for 30 min at 4℃ in cold PBS with 1% BSA and 0.01% NaN₃. The cells were then permeabilised with 0.1% Triton in PBS with 1% BSA and stained with 10 μg/ml anti-CD3 epsilon mAb (UCHT1, Biolegend) followed by anti-mouse IgG1-AF546 and coverslips mounted with Prolong Gold anti-fade (ThermoFisher). Images were acquired using the DeltaVision ELITE Image Restoration Microscope (Applied Precision) coupled to an inverted Olympus IX71 microscope and a CoolSNAP HQ2 camera, deconvolved with softWoRx 5.0 and processed using Huygens Professional v4.0 and Adobe Photoshop CC 2018.

**Metabolic assay.** CD8+ T cells were isolated from fresh PBMC using a positive CD8+ T cell isolation kit (Miltenyi Biotech) according to the manufacturer's instructions. Isolated CD8+ T cells were cultured in cRPMI+10% FBS+ 20 U/ml IL-2 at 37℃ and treated with 1 μM Avasimibe or equivalent concentration of DMSO. After 16 h, CD8+ T cells were counted, viability was confirmed and cells were stimulated with 50 ng/ml PMA and 500 ng/ml Ionomycin (Sigma-Aldrich) in Seahorse XF media (Agilent) to measure OCR and ECAR in real-time using a Seahorse XFe96 Analyzer (Agilent). After establishment of a stable baseline OCR,

we first administered oligomycin (ATP synthase/complex V inhibitor), followed by mitochondrial uncoupling with carbonyl cyanide-4-(trifluoromethoxy)phenylhydrazone (FCCP) and rotenone and antimycin A (Rot/AA; complex I and III inhibitors) according to the manufacturer's instructions (all compounds from Agilent). Calculation of basal respiration (baseline OCR minus OCR after Rot/AA), ATP production (baseline OCR minus OCR after Oligomycin) and maximal respiration (OCR after FCCP minus OCR after Rot/AA). All experiments were performed in five replicates for each donor.

**HBV replication models.** HepG2-NTCP cells (kindly provided by Stephan Urban, Heidelberg) were transduced with Ad-HBV (multiplicity of infection, MOI 20) or infected with HBV (MOI 200) and incubated with 2.5 μg/ml and 5 μg/ml of the ACAT inhibitor Avasimibe or 1 μM of the NUC Entecavir (ETV, Sigma-Aldrich), which were replenished every 3d. Drugs were applied for a total of 6d either at the start of infection or 6d after the infection as indicated. Extracellular HBV DNA, cccDNA, and pgRNA were quantified by quantitiative PCR (qPCR) as previously described[68]; HBsAg and HBeAg were quantified by ELISA (Autobio Diagnostics). Viability of HepG2-NTCP was assessed using a WST-8 Cell Proliferation Kit (Cayman Chemical) according to the manufacturer's instructions.

**HBV uptake assay.** HepG2-NTCP cells pre-treated with 2.5% DMSO for 3d were treated with 5 μg/ml Avasimibe or 400 nM MyrB for 1 h prior to infection. Cells were chilled on ice for 15 min before infection with HBV (MOI 200) and maintained at 4℃ for 1 h to permit viral attachment. Cells were subsequently transferred to 37℃ for 6 h after which non-internalised virus was removed by treating with trypsin for 3 min followed by three washes with PBS. The cells were harvested and HBV DNA measured by qPCR. Drug treatment was maintained for the duration of the experiments.

**Gene expression analysis.** To examine the gene expression of SOAT1 (ACAT1) and SOAT2 (ACAT2) in T cells, we probed previously published sc RNA-sequencing data from human blood, liver and HCC tissue focusing on CD8+ and CD4+CD25− T cells[20] (available on Gene Expression Omnibus, GSE98638).

To examine gene expression of SOAT1 and SOAT2 in CHB, we probed previously published bulk liver microarray data from three different data sets[48–50] (available on Gene Expression Omnibus, GSE83148; GSE84044; GSE65359). Where gene expression was quantified through multiple probes, the probe with the highest median expression was interrogated.

**Statistical analysis.** Statistical analyses were performed with Prism 7.0 (GraphPad) as indicated in figure legends (Wilcoxon matched-pairs signed-rank test, Mann–Whitney test, Friedman test with Dunn's multiple comparisons test, Kruskal–Wallis test with Dunn's multiple comparisons test, Fisher's exact test, Spearman correlation) in raw data with significant differences marked on all figures. In experiments with a sample size of $n < 50$ non-normal distribution was assumed and non-parametric statistical analysis was chosen. In experiments with a sample size of $n \geq 50$, normality was assessed by D'Agostino–Pearson normality test prior to further analysis and the following statistical tests were chosen accordingly as indicated in the figure legends. All tests were performed as two-tailed tests, and for all tests, significance levels were defined as not significant (ns) $P \geq 0.05$; *$P < 0.05$; **$P < 0.01$; ***$P < 0.001$; ****$P < 0.0001$.

**Reporting summary.** Further information on research design is available in the Nature Research Reporting Summary linked to this article.

## Data availability

Gene expression analysis was performed using previously published data sets publicly available on Gene Expression Omnibus (GSE98638, GSE83148, GSE84044, GSE65359). All other data that support the findings of this study are available from the corresponding author upon reasonable request. Source data are provided with this paper.

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

## Acknowledgements

We are grateful to patients participating in the study and to clinical staff who helped with participant recruitment and sample acquisition, including NHS transplant coordinators, research nurses and the Tissue Access for Patient Benefit (TapB) project at The Royal Free Hospital (RFH) London funded by RFH Charity and UCLH/UCL BRC. This work was funded by a Wellcome Trust Investigator Award (214191/Z/18/Z) to M.K.M., Cancer Research UK project grant (26603) to M.K.M., T.M. and H.J.S., Cancer Research UK HUNTER: Hepatocellular Carcinoma Expediter Network to M.K.M. and T.M., a Wellcome Trust Investigator Award (200838/Z/16/Z) to J.A.M., an MRC project grant (MR/R022011/1) to J.A.M., and a DFG Research Fellowship (SCHM3310/1-1) to N.M.S.

## Author contributions

N.M.S., J.A.M. and M.K.M. conceived the project; N.M.S., P.A.W., L.J.P., E.C.J., C.J., J.A.M., M.K.M. designed experiments; N.M.S., P.A.W., M.O.D., C.J. generated data; N.M.S., P.A.W., L.J.P., M.O.D., J.M.H., J.G., H.J.S., I.P.-T., C.J., E.C.J., J.A.M. and M.K.M. analysed and interpreted data; N.M.S., L.J.P., A.R.B., L.S., A.J.-S., N.Z., O.E.A., S.K., M.H.H., B.D., T.M. and M.K.M. provided or processed essential patient samples and clinical data and/or essential tools. N.M.S. and M.K.M. prepared the manuscript. All authors provided a critical review of the manuscript.

## Competing interests

The work described in this manuscript (combined immune-boosting and antiviral effects of ACAT inhibition in HBV and HBV-related HCC) is protected under an international patent application no.1917498.6 entitled Treatment of HBV Infection filed by applicant UCL Business Ltd. on 29 November 2019 with inventors named as M.K. Maini, N. Schmidt, A. Burton, P. Wing and J. McKeating. Unrelated to the content of this manuscript, the Maini laboratory has received unrestricted funding from Gilead, Roche and Immunocore. M.K.M. has sat on advisory boards/provided consultancy for Gilead Sciences, Hoffmann La Roche, Galapagos NV, GSK, and Freeline, with no funds being taken personally. L.J.P. has provided consultancy for Gilead Sciences. H.J.S. is a founder of Quell Therapeutics, shareholder of Quell Therapeutics, shareholder of Kuur Therapeutics and scientific advisor for PanCancerT. All remaining authors declare no competing interests.
