## [Peer Review File · Nature Communications]

REVIEWER COMMENTS

Reviewer #1 (Remarks to the Author):

Schmidt et al describe in this paper the effect of ACAT inhibition on the function human primary CD8 T cells in an antigen-specific manner. They show that ACAT inhibition boost cytokine production, decrease lipid droplets, increase lipid raft content, TCR downstream signalling phosphorylation events and metabolic reprogramming.

This is the first time that inhibition of cholesterol esterification is related to anti-viral and anti-tumour response in primary human cells; hence, this work is relevant not only because it expands mouse models' data to primary human cells, as the authors' reference (Yang et al, Nature 2016), but also because it provides more evidence that supports the exploration of new therapeutic avenues that employ drugs that target cholesterol metabolism in infection and cancer. Moreover, their data suggests that ACAT inhibition might overcome a major hurdle in the quest to use metabolic reprogramming to treat human disease: while most of current metabolic checkpoints have similar effect in immune cells and their targets, ACAT inhibition seems to boost immune function while can be detrimental for tumour or infected cells.

I have some suggestions that, in my opinion, can improve the manuscript:

1. The description of the experiments and the culture system used, which is key to understand the results, it is not very obvious. There are important changes in the protocols (i.e. use of different ACAT inhibitors, 8 days vs 16h exposure to inhibition) within figure, which difficult the interpretation of the data. I would encourage authors to explain which inhibitor has been used and for how long in both results and each figure legend for clarity purposes. I have assumed that PBMC (which, as stated in the methods section, can be activated either for 16h or 8d) have been activated for 8d in all experiments where cells are activated with HBV-OLP (see line 444) and that "untreated" (i.e. fig2a), "peptide alone" (i.e. fig. 1b) or "stim" (fig. 2e) are also diluent (DMSO in all cases??) control.

2. ACAT inhibition:

- The paper uses 2 different ACAT inhibitors (K604 and Avasimibe) in different experiments. PBMC are exposed to Avasimibe for 8 days, while IHL, TIL and PBMC are exposed to K604 for 16h. While it is reassuring that both inhibitors cause similar phenotypes in their respective experiments, a side to side comparison of both would make findings more robust.
- The work provides some evidence as to how the inhibitor might work (it decreases LipidTOX staining, while increasing CTB). However, this is shown only in PBMC culture condition, which exposes the cells to the drug (Avasimibe) for 8 days. The mechanism by which ACAT inhibition modulates CD8 T cell function may well be different when cells are exposed for shorter time, or to a different drug. Authors should either do side to side comparison or at least note this in the discussion.
- Could authors explain why they use these ACAT inhibitor concentrations? Have they done any titration? In the case of K604, the IC50 for ACAT-1 (likely to be the isoform expressed by CD8 T cells) is 0.45uM (and 0.07uM for human monocyte cell lines) (see Ikenoya et al Atherosclerosis 2007), while Avasimibe IC50 is 3.3uM (Selleckchem webpage).

3. Statistical methodology:

- Authors use non-parametric test (Wilcoxon matched-pairs) possibly due to the non-normal distribution of the data (or the low n in some cases, or both). However, the methods do not

describe how normality has been assessed. A complete description of the statistical methodology is encouraged.

- A statement ensuring that statistics shown in normalised data have been performed in raw data is missing.
- If distribution is not normal, then mean should not be used as a descriptive, but median.

4. Effect of ACAT inhibition on CD8 effector function:

- Due to the low frequency of antigen specific cells in periphery, the frequency of responding cells is very low in some patients (for instance, fig1a shows a 0.6% background response, which is above the level of 6 patients' raw data). It would be interesting to know whether inhibition has the same effect on polyclonal stimulated cells. This will also help understanding SFig1F, where levels of IFN γ + cells in unstimulated cells are generally higher than those activated with peptide (SFig1C).
- Even though the authors highlight the heterogenous response within their patients, only half of the cohort (13 maybe?) show a substantial increase in IFN γ production when compared to solvent control. The use of a second IFN γ readout might help to make this finding more robust. Maybe ELISA will work here: I am aware that the culture contains bulk PBMC hence the secretion of IFN γ could not be definitely ascribed specifically to CD8 T cells, but Yang et al show that the ACAT-1 inhibitory effect targets CD8 T cells and not CD4 T cells (possibly the second main producers of cytokine in their system after 8 days of culture). CD4 cells might also be less responsive to their peptide combinations?

5. Addressing heterogeneity of response

It is a fact that human immune response is heterogeneous. This is a constant challenge when studying primary human cells and I really praise the authors for this effort, as it is extremely difficult to address. However, this heterogeneity on the response is not consistent. For instance, Fig1d show the CFSE profile of 8 patients (not 7, as stated in the legend) that increase their proliferative state in the presence of Avasimibe. Why is this response so homogeneous when compared to the cytokine response? This is also relevant for lipid accumulation data: figure 2A shows a consistent downregulation on LipidTOX signal for 7/8 patients, despite heterogenous cytokine production. Could authors discuss this?

6. ACAT inhibition mechanism

Authors describe a double possible mechanism whereby ACAT inhibition promotes CD8 T cell function: the formation of lipid rafts that facilitate TCR signalling and also a metabolic switch towards OXPHOS.

- The data shown here do not demonstrate a direct relation between increased lipid microdomains and increased pAKT, pErk and pS6 upon ACAT inhibition, as the authors state. The experiments might benefit of some controls. For instance, would cholesterol supplementation mimic the effect of ACAT inhibition? Or a negative control: would cholesterol extraction from the membrane (with methyl- β -cyclodextrin or 7-ketocholesterol) revert the effect of ACAT inhibition. These controls would support the idea that the effector phenotype is linked to increased cholesterol content in the plasma membrane that facilitate TCR signalling.
- Decreased MFI in LipidTOX signal can also be explained by a general downsize of the cell. Could authors check that this is not the case by showing no changes on cell size by checking MFI changes on FSc, CD3 and/or CD8 staining?
- CTB has been disputed as a lipid raft marker in some cases, as the authors themselves have reported (Miguel J Immunol 2011). The use of more specific dyes such as di-4-ANEPPDHQ might give a better picture on whether ACAT inhibition changes or not the fluidity of the plasma membrane.

7. Co-operative effects between ACAT and PD-1 inhibition.

The data presented here shows mainly associations between the expression of surface markers, rather than exploring a possible role for ACAT inhibition and check point markers expression on boosting IFN γ expression and effector CD8 T cell function. Although the data is interesting and antagonises some of the previous work by Ma et al (Cell Metabolism 2019), I am unsure how this contributes to the overall message of the work.

- I disagree in the data interpretation: authors describe in fig3A that PD1+ CD8 T cells showed marked lower levels of CTB staining. However, the data show that there is a reduced frequency of PD-1+ cells in the CTBh+ compartment, which is different. Should the authors want to assess the levels of CTB in PD-1+ cells, then they should gate that population and check for differences in CTB MFI between PD-1+ and PD-1- cells. This interpretation is followed in fig3C, where intensity of expression and frequency of CTB+ cells do not match between CD28+ and CD28- cells (CD28- cells have lower proportion, but they have much brighter expression of CTB, as seen in the representative data).

- When addressing the frequency of responders to the combo condition, could authors please clarify how being a responder has been defined? As in some cases, the difference between one condition and the other looks really minimal. The conclusion of the experiment is over simplified, as the combo only works on some patients and there is no mechanistic link or co-variate that explains this heterogeneity.

I also would request clarification on the following minor issues:

- The title of the work could be improved to reflect the data more accurately, for example, including the fact that all is human work and specifically CD8 T cells.
- Introduction might benefit from a specific description of the work that has already been done in the field of ACAT inhibition. For instance, a brief but clear description that ACAT inhibition role in modulating CD8 T cell responses in the tumour setting is controversial (Yang et al Nature 2016 and Ma et al CellMetab 2019).
- Fig1C shows a possible gender bias in the data. However, the authors do not follow this up, neither comment it in the discussion. It is an interesting observation, but lacks place in the main figure of the paper without a follow up. Similar comment for fig1G, where authors do not follow up on this interesting observation, neither they discuss why this might (or not) be relevant.
- Fig1F – what peptide pool is these data referring to? Legend indicates that data shows HBV OLP +/- inhibitor compared to unstimulated control. Do authors mean normalised to unstimulated control? And compared with and without inhibitor, as the graph seems to suggest? This is confusing (please also refer to my comment number 1)
- Fig4D shows the results of only one experiment (representative of two) with a really modest increase in IFN γ expression. I do not think the data supports the conclusion. More repeats and/or another readout for IFN γ production would be required.
- Definitions of HBc (core) and HBs are missing (they are probably very familiar to hepatitis B experts, but not to the general audience). A table with the peptides used might also be informative.

Espe

Esperanza Perucha, PhD
Lecturer in Experimental Rheumatology
King's College London

Reviewer #2 (Remarks to the Author):

Lipid metabolism in antiviral/antitumor immunity is an evolving field. ACAT, the cholesterol esterification enzyme, is known to regulate both HCC cells and CD8 T cells in mouse tumor models. In the manuscript "Targeting Acyl-CoA:cholesterol acyltransferase as a dual viral and T-cell metabolic checkpoint", Schmidt et al. studied the effect of ACAT inhibitor on patient-derived anti-HBV and anti-HCC T cells, and also on HBV particle genesis. Although the mechanism part to some extent is not of high novelty, the ex vivo studies with HBV/HCC patient T cells make this work highly interesting and valuable for developing new therapeutics against HBV and HBV-related HCC. I only have a few minor comments.

1. In addition to use measure raft-associated cholesterol enriched by CTB, the authors shall use the ALOD4 probe to measure whether free cholesterol level of the plasma membrane could be influenced by ACAT inhibition.
2. For the HCC TIL analysis, why in some patients ACAT inhibition induced contrary effect under different TAA stimulation?
3. Figure 4D, 4E and S4D lack statistical analysis.

Reviewer #3 (Remarks to the Author):

In this paper, authors found that acyl-CoA:cholesterol acyltransferase (ACAT) inhibition reduced CD8+T cell neutral lipid droplets and promoted lipid microdomains, enhancing TCR signalling and TCR-independent bioenergetics. Dysfunctional HBV- and HCC-specific CD8+T cells were rescued by ACAT inhibitors directly ex vivo from human liver and tumour tissue respectively, including tissue resident responses. ACAT inhibition enhanced in vitro responsiveness of HBV-specific CD8+T cells to PD-1 blockade and increased the functional avidity of TCR-gene-modified T cells. Finally, ACAT regulated HBV particle genesis, with inhibitors reducing both virions and subviral particles. Thus, authors proposed that ACAT inhibition provided a paradigm of a metabolic checkpoint that is able to constrain tumours and viruses but rescue exhausted T cells, rendering it an attractive therapeutic target for the functional cure of HBV and HBV-related HCC. However, the evidence is inadequate to support this conclusion, and some of the data required more experiments to verify since the difference between experiment groups was not significant. Statistical analysis should be shown in the figure legend of figures and supplementary figures.

1. Inhibition of SOAT1 (also known as ACAT) markedly reduced the size of tumours that had high levels of SOAT1 expression. In this study, authors described ACAT a dual viral and T-cell metabolic checkpoint. More evidence is required to clear explain the mechanisms of ACAT behaviour in immune cells in addition to the tumor cells.
2. How about the ACAT metabolism pathway in intrahepatic CD8+T cells during chronic HBV infection? And what are the main factors derived from HBV or HCC that are responsible for the altered ACAT? These are important for explore the mechanisms of ACAT in CD8+T cells.
3. To demonstrate ACAT as an attractive therapeutic target for the functional cure of HBV and HBV-related HCC, experiments in vivo need to be under investigation. For example, HBV-carrier mouse model, acute HBV infection mouse model, and HBV-related HCC mouse

model may be used to observe the effects of ACAT inhibition in vivo.

4. IFN- γ secretion is only one of the functional molecules of CD8+T cells. More work need to be performed to explore the mechanisms of ACAT inhibition in CD8+T cells.

5. HCC related antigens were described. How to select these antigens? It was wonder whether they are highly expressed in HCC tumor cells from the patients?

6. As shown in figures, the frequency of IFN- γ +CD8+T cells were still in a low level in the groups of ACAT inhibition. What about the absolute numbers of IFN- γ +CD8+T cells? The effects of these increased CD8+T cells during the liver injury should be investigated further.

7. In supplemental figure 4, the frequency of intratumoral CD3+T cells was very low. How about their absolute numbers? Further, intratumoral CD8+T cells deserved investigating for their function with ACAT inhibition or not.

8. In this study, authors determined the ACAT metabolism in CD8+T cells. During the development of HCC, other immune cells such as NK cells, NKT cells, $\gamma\delta$ T cells, which were predominant cell populations in the liver, played critical anti-tumor activities. Does ACAT inhibition affect their functions?

Changes are detailed in black font in the point-by-point response below and highlighted in red in the revised manuscript.

Reviewer 1:

.... This is the first time that inhibition of cholesterol esterification is related to anti-viral and anti-tumour response in primary human cells; hence, this work is relevant not only because it expands mouse models' data to primary human cells, as the authors' reference (Yang et al, Nature 2016), but also because it provides more evidence that supports the exploration of new therapeutic avenues that employ drugs that target cholesterol metabolism in infection and cancer. Moreover, their data suggests that ACAT inhibition might overcome a major hurdle in the quest to use metabolic reprogramming to treat human disease: while most of current metabolic checkpoints have similar effect in immune cells and their targets, ACAT inhibition seems to boost immune function while can be detrimental for tumour or infected cells.

We appreciate the reviewer appraising the biological and translational advances of our study so positively.

1. The description of the experiments and the culture system used, which is key to understand the results, it is not very obvious. There are important changes in the protocols (i.e. use of different ACAT inhibitors, 8 days vs 16h exposure to inhibition) within figure, which difficult the interpretation of the data. I would encourage authors to explain which inhibitor has been used and for how long in both results and each figure legend for clarity purposes. I have assumed that PBMC (which, as stated in the methods section, can be activated either for 16h or 8d) have been activated for 8d in all experiments where cells are activated with HBV-OLP (see line 444) and that "untreated" (i.e. fig2a), "peptide alone" (i.e. fig. 1b) or "stim" (fig. 2e) are also diluent (DMSO in all cases??) control.

We thank the reviewer for pointing out the parts of our study protocol needing clarification. We have extended the relevant methods sections and have amended each figure legend to clarify the ACAT inhibitor used and the duration of treatment for each individual experiment. An equivalent concentration of the solvent DMSO was added to all conditions without ACAT inhibition and we have now included this information in all figure legends.

2. ACAT inhibition:

- The paper uses 2 different ACAT inhibitors (K604 and Avasimibe) in different experiments. PBMC are exposed to Avasimibe for 8 days, while IHL, TIL and PBMC are exposed to K604 for 16h. While it is reassuring that both inhibitors cause similar phenotypes in their respective experiments, a side to side comparison of both would make findings more robust.

To show that the enhancement of CD8⁺T-cell function is a generalized effect of ACAT inhibition rather than that of one specific compound, we used two different ACAT inhibitors in this study (Avasimibe and K-604). We have now added a side-by-side comparison of the two ACAT inhibitors, showing similar enhancement of CD8⁺T-cell proliferation. These data have now been added to the manuscript (SuppFig 1g and results section p6, line 104-106).

- The work provides some evidence as to how the inhibitor might work (it decreases LipidTOX staining, while increasing CTB). However, this is shown only in PBMC culture condition, which exposes the cells to the drug (Avasimibe) for 8 days. The mechanism by which ACAT inhibition modulates CD8 T cell function may well be different when cells are exposed for shorter time, or to a different drug. Authors should either do side to side comparison or at least note this in the discussion.

We thank the reviewer for this important suggestion and agree that timing is crucial in the evaluation of metabolic changes. We therefore performed a time course and indeed saw a reduction of neutral lipid droplets as early as 1h after treatment with ACAT inhibition. We have included the new data showing effects at 1h in the main Fig 2a and effects maintained after repeated 48hr dosing for 7 days in SuppFig 2a, and clarified how early these changes start in results p8, line 156-158.

- Could authors explain why they use these ACAT inhibitor concentrations? Have they done any titration? In the case of K604, the IC50 for ACAT-1 (likely to be the isoform expressed by CD8 T cells) is 0.45uM (and 0.07uM for human monocyte cell lines) (see Ikenoya et al Atherosclerosis 2007), while Avasimibe IC50 is 3.3uM (Selleckchem webpage). In our initial experiments, the concentration of Avasimibe used in previous publications (e.g. Yang et al, Nature 2016) showed high cell toxicity; we therefore titrated the dose down to reduce toxicity and enhance CD8⁺T-cell expansion. We further evaluated different *in vitro* dosing strategies and saw the most efficient enhancement of CD8⁺T-cell function when the ACAT inhibitor was given every second day throughout an 8d cell culture to allow low frequency HBV-specific T-cells to expand.

3. Statistical methodology:

- Authors use non-parametric test (Wilcoxon matched-pairs) possibly due to the non-normal distribution of the data (or the low n in some cases, or both). However, the methods do not describe how normality has been assessed. A complete description of the statistical methodology is encouraged.

- A statement ensuring that statistics shown in normalised data have been performed in raw data is missing.
- If distribution is not normal, then mean should not be used as a descriptive, but median.

Due to the relatively low n numbers (<50) we did not assess normality as normality tests in small sample sizes have low power to detect Gaussian distribution. Choosing a parametric test for samples with non- Gaussian distribution can overestimate statistical significance and we therefore used non-parametric tests in all experiments. Following the valid comment by the reviewer that the mean should not be used as a descriptive in non-normally distributed data, we have removed the mean from all figures, simply displaying individual data points for all paired samples instead.

In response to another reviewer's comment, we have added scRNA-sequencing data to the revised manuscript with n>50. We performed a D'Agostino & Pearson normality test for these samples and as the samples did not pass the normality test we have again applied non-parametric statistical tests and displayed the median not mean. We have clarified the statistical methodology in the relevant Methods section (p28, lines 638-647).

4. Effect of ACAT inhibition on CD8 effector function:

- Due to the low frequency of antigen specific cells in periphery, the frequency of responding cells is very low in some patients (for instance, fig1a shows a 0.6% background response, which is above the level of 6 patients' raw data). It would be interesting to know whether inhibition has the same effect on polyclonal stimulated cells. This will also help understanding SFig1F, where levels of IFN γ + cells in unstimulated cells are generally higher than those activated with peptide (SFig1C).

In response to this point it is important to clarify that all data showing the percentage of cytokine producing T-cells after HBV or tumour peptide stimulation were presented after subtraction of the matching unstimulated background (including e.g. Fig1a and SuppFig1c). Instead, raw data in representative flow cytometry plots show the background unstimulated response and the peptide responses with or without ACAT inhibition *before* subtraction of background (as in the example referred to, where the 0.6% background response depicted was subtracted from the 1.3% response shown to obtain the peptide-specific response of 0.7% plotted in the summary histogram). We previously explained this approach, widely used in antigen-specific T-cell studies, in the Methods section, but we have now also clarified in each relevant figure legend when background unstimulated wells have been subtracted to obtain peptide-specific responses. The only exception to this approach was SuppFig 1h (previously SuppFig 1f), where there is no peptide stimulation and background responses with or without ACAT inhibition are therefore presented.

As requested by the reviewer, we have also now added data using polyclonal (anti-CD3/antiCD28mAb) stimulation of TILs freshly isolated from tumour tissue. Global tumour-infiltrating CD8⁺T-cells, many of which have been shown in other studies to be directed against tumour neoantigens, showed increased IFN γ production in 6 out of 7 patients after treatment with ACAT inhibition (new SuppFig 4j, results p13 line 284-285). In contrast, ACAT inhibition did not boost the highly functional circulating T-cell response to the well-controlled virus CMV (new SuppFig 1j, results p6 line 107-109).

- Even though the authors highlight the heterogenous response within their patients, only half of the cohort (13 maybe?) show a substantial increase in IFN γ production when compared to solvent control. The use of a second IFN γ readout might help to make this finding more robust. Maybe ELISA will work here: I am aware that the culture contains bulk PBMC hence the secretion of IFN γ could not be definitely ascribed specifically to CD8 T cells, but Yang et al show that the ACAT-1 inhibitory effect targets CD8 T cells and not CD4 T cells (possibly the second main producers of cytokine in their system after 8 days of culture). CD4 cells might also be less responsive to their peptide combinations? This type of variable response to immunotherapeutic strategies is typical of other published studies on HBV and tumour-specific T-cells, including for strategies like PD-1 blockade that showed equivalent or less consistent rescue *in vitro* for T-cells specific for HBV (*Boni J Virol 2007*) and HIV (*Day Nature 2006*) but has progressed into clinical trials in these settings. *In vitro* studies will tend to underestimate therapeutic responses achieved *in vivo*, but even *in vivo* responses are typically restricted to subsets of cohorts in chronic viral infections and tumours due to the multi-faceted nature of T-cell constraints in these diseases (new discussion p19 line 431-433). As we show in Fig.3e+f, the proportion of responders is actually higher for ACAT inhibition with or without PD-1 blockade than for PD-1 blockade alone, whilst ACAT rescue of TILs is highly consistent, suggesting this will be an advance on the existing limited response to PD-1 blockade in HCC clinical trials. In HBV, the number of responders to ACAT inhibition and the degree of rescue in the blood is less striking we agree, but still significant for the cohort and again better than many published immunotherapeutic strategies for these notoriously low-frequency T-cells, whilst importantly, we show a very consistent boosting in the liver.

Regarding the suggestion to use supernatant ELISA, this would be too inaccurate to interpret when evaluating mixed cultures with such low frequency responses, particularly because we have found that CD4⁺T-cells do also respond to ACAT inhibition. We have now included data confirming that a similar proportion of human CD4⁺T-cells transcribe ACAT1 compared to CD8⁺T-cells (new SuppFig 1k+l and SuppFig4 b+c; results p6 lines 112-116 and p12 lines 267-270, respectively). In contrast to previously published data in a tumour mouse model with ACAT knockout (Yang et al, *Nature 2016*), ACAT inhibition increased human IFN γ + HBV- and TAA-specific CD4⁺T-cells isolated from the liver and tumour, respectively, with a trend to increased HBV-specific IFN γ production by CD4⁺T-cells in the blood; all these CD4⁺ rescue data have now been added to the manuscript (SuppFig 1f,r,s and SuppFig 4i, results p5 line 98-99; p6 line 126-128; p7 line 130-132; p13 line 283-284). Instead of the suggested ELISA of mixed cultures, we have added Luminex analysis of supernatant soluble mediators in the section evaluating ACAT inhibition in pure populations of TCR-transduced T-cells (new SuppFig 4m, results p13-14 lines 300-306).

5. Addressing heterogeneity of response

It is a fact that human immune response is heterogeneous. This is a constant challenge when studying primary human cells and I really praise the authors for this effort, as it is extremely difficult to address. However, this heterogeneity on the response is not consistent. For instance, Fig1d show the CFSE profile of 8 patients (not 7, as stated in the legend) that increase their proliferative state in the presence of Avasimibe. Why is this response so homogeneous when compared to the cytokine response? This is also relevant for lipid accumulation data: figure 2A shows a consistent downregulation on LipidTOX signal for 7/8 patients, despite heterogenous cytokine production. Could authors discuss this?

We are grateful to the reviewer for prompting us to clarify this distinction. The reason the rescue of proliferation is more consistent than that of IFN γ is because the former was only analysed on a pre-selected group of known responders by the HBV-specific IFN γ assay, since we required an IFN γ response to analyse CFSE dilution within this antigen-specific fraction (now explained in Results p5 line 103). On reviewing these data, we have decided to remove two outlier data points from Fig.1d due to technical issues with poor CFSE staining but the summary data remain homogenous and statistically significant.

The LipidTOX staining is more consistent because it was measured on global unstimulated CD8⁺T-cells so is not dependent on the heterogenous and subtle nature of low-frequency antigen-specific T-cell rescue discussed above.

6. ACAT inhibition mechanism

Authors describe a double possible mechanism whereby ACAT inhibition promotes CD8 T cell function: the formation of lipid rafts that facilitate TCR signalling and also a metabolic switch towards OXPHOS.

- The data shown here do not demonstrate a direct relation between increased lipid microdomains and increased pAKT, pErk and pS6 upon ACAT inhibition, as the authors state. The experiments might benefit of some controls. For instance, would cholesterol supplementation mimic the effect of ACAT inhibition? Or a negative control: would cholesterol extraction from the membrane (with methyl- β -cyclodextrin or 7-ketocholesterol) revert the effect of ACAT inhibition. These controls would support the idea that the effector phenotype is linked to increased cholesterol content in the plasma membrane that facilitate TCR signalling.

We agree with the reviewer; our data suggest that a combination of metabolic changes, including increased lipid microdomains, reduced lipid droplets and altered bioenergetics (OXPHOS, glycolysis) could all contribute to the observed enhanced TCR signalling and increased T-cell function. We have therefore amended this section to clarify that the increased T-cell signalling we observed upon TCR stimulation likely reflects extensive metabolic re-programming rather than necessarily just resulting from enhanced lipid microdomains (results p8 line 160-161 and p9, line 181).

In response to this reviewer's suggestion we have also now evaluated the efficacy of ACAT inhibition in the presence of high cholesterol media. CD8⁺T-cells exposed to high cholesterol media showed a significantly enhanced proliferative response to ACAT inhibition compared to those in normal media (new SuppFig 1t, results p7 lines 133-137). This is a potential explanation for the enhanced response of liver-infiltrating T-cells compared to PBMC and supports the efficacy of ACAT inhibition in high cholesterol environments like the liver and tumours (discussion p19 lines 436-439).

- Decreased MFI in LipidTOX signal can also be explained by a general downsize of the cell. Could authors check that this is not the case by showing no changes on cell size by checking MFI changes on FSc, CD3 and/or CD8 staining?

ACAT inhibition did not lead to changes in cell size as assessed by FSC. We have added these data to the manuscript (Supp Fig 2b, results p8 line 158).

- CTB has been disputed as a lipid raft marker in some cases, as the authors themselves have reported (Miguel J Immunol 2011). The use of more specific dyes such as di-4-ANEPPDHQ might give a better picture on whether ACAT inhibition changes or not the fluidity of the plasma membrane.

We tried using di-4-ANEPPDHQ to analyse membrane lipids but did not observe any changes of generalized polarisation (GP, a normalized intensity ratio of the two different spectral channels of di-4-ANEPPDHQ) after treatment of unstimulated PBMC with ACAT inhibition. However, there were technical issues that limit our confidence in the results of this assay, including not being able to test it on CHB samples (that require fixation which interferes with this assay) and non-specific changes in di-4-ANEPPDHQ polarisation that were found to be induced by our T-cell stimulation protocol. We have instead included some additional data obtained using Filipin complex from *Streptomyces filipinensis* to detect free membrane cholesterol. We observed a significant increase of free membrane cholesterol after treatment with ACAT inhibition in high cholesterol media, further supporting the efficacy of ACAT inhibition in high cholesterol environments such as the liver (new SuppFig2d, results p8 line 161-165, discussion p19 lines 436-439).

7. Co-operative effects between ACAT and PD-1 inhibition.

The data presented here shows mainly associations between the expression of surface markers, rather than exploring a possible role for ACAT inhibition and check point markers expression on boosting IFN γ expression and effector CD8 T cell function. Although the data is interesting and antagonises some of the previous work by Ma et al (Cell Metabolism 2019), I am unsure how this contributes to the overall message of the work.

- I disagree in the data interpretation: authors describe in fig3A that PD1⁺ CD8 T cells showed marked lower levels of CTB staining. However, the data show that there is a reduced frequency of PD-1⁺ cells in the CTB⁺ compartment, which is different. Should the authors want to assess the levels of CTB in PD-1⁺ cells, then they should gate that population and check for differences in CTB MFI between PD-1⁺ and PD-1⁻ cells. This interpretation is followed in fig3C, where intensity of expression and frequency of CTB⁺ cells do not match between CD28⁺ and CD28⁻ cells (CD28⁻ cells have lower proportion, but they have much brighter expression of CTB, as seen in the representative data).

Apologies that our gating strategy was not clear; we did in fact pre-gate as suggested on PD-1⁺ and PD-1⁻ CD8⁺T-cells and then examine CTB staining for each fraction. To clarify this, we have included sample FACS plots showing gating on PD-1⁺ vs PD-1⁻ arrowed to their respective CTB expression in Fig3a. In response to the reviewer's request, we have also now included an extra figure (SuppFig 3a) showing that overall CTB MFI (as well as %CTB⁺) is lower in the PD-1⁺ fraction. Similarly, we have inserted the equivalent illustrative gating plots for CD28⁺/⁻ in Fig3c and included MFI summary data (new SuppFig 3b). Although, as the reviewer correctly observes, there is a subpopulation with high MFI of CTB within the CD28⁻ fraction, overall both the %CTB⁺ (Fig3c) and the total CTB MFI (new SuppFig 3b) are still significantly higher in the CD28⁺ fraction.

- When addressing the frequency of responders to the combo condition, could authors please clarify how being a responder has been defined? As in some cases, the difference between one condition and the other looks really

minimal. The conclusion of the experiment is over simplified, as the combo only works on some patients and there is no mechanistic link or co-variate that explains this heterogeneity.

Treatment response was defined as increased or *de novo* IFN γ production. In addition to the methods section, we have now included this information in the relevant figure legends for further clarification. As reported for other *in vitro* and *in vivo* immunotherapeutic strategies e.g. PD-1 blockade, responses to *in vitro* ACAT inhibition and a combination of ACAT inhibition and PD-1 blockade are heterogenous. ACAT inhibition and PD-1 blockade given together was additive in 11 out of 26 patients, in whom the strongest boosting of HBV-specific responses was seen with the combination (example flow cytometry plots and red boxed patients in summary below the histogram, Figure 3e). The addition of ACAT inhibition to PD-1 blockade also significantly enhanced the fold increase in HBV-specific CD8⁺T-cells for the cohort compared to PD-1 blockade alone; to strengthen this important finding we have moved this figure from the supplementary to main figure (new Fig3f). Please also see response to point 4 above regarding our more nuanced interpretation of these changes in the context of similar published work on *in vitro* rescue of virus and tumour-specific T-cell responses.

I also would request clarification on the following minor issues:

- The title of the work could be improved to reflect the data more accurately, for example, including the fact that all is human work and specifically CD8 T cells.

We have added 'human' to the title. We have left it as 'T-cells' since we now include CD4⁺ as well as CD8⁺T-cell data.

- Introduction might benefit from a specific description of the work that has already been done in the field of ACAT inhibition. For instance, a brief but clear description that ACAT inhibition role in modulating CD8 T cell responses in the tumour setting is controversial (Yang et al Nature 2016 and Ma et al CellMetab 2019).

We are not aware of any controversy in the literature regarding the effect of ACAT inhibition on T-cells. We introduced the key murine study on this (Yang et al Nature 2016) in the introduction and have added a new paper on the application of the ACAT inhibitor Avasimibe to optimise CAR-T-cells using click chemistry (Hao et al, Science Trans Med 2020) in the Discussion (p20 line 445-446). We agree that the Ma et al Cell Metab 2019 paper raises an important differential effect of high cholesterol in the tumour microenvironment on tumour T-cell PD-1 expression (outside the context of ACAT inhibition), so have now expanded the results and discussion to stress that ACAT inhibition does not increase T-cell PD-1 expression and is even more effective in a high cholesterol environment (SuppFig 4a, new SuppFig 1t and SuppFig 2d; results p7 line 133-137, p8 line 161-165, p12 line 262-267; discussion p19 line 436-439).

- Fig1C shows a possible gender bias in the data. However, the authors do not follow this up, neither comment it in the discussion. It is an interesting observation, but lacks place in the main figure of the paper without a follow up. Similar comment for fig1G, where authors do not follow up on this interesting observation, neither they discuss why this might (or not) be relevant.

We agree that future studies are needed to understand the mechanism underlying the gender bias in responsiveness and have now added to the discussion that this could be used as a potential selection biomarker for this therapy if it holds out in bigger studies (Discussion p19 line 431-433 and 438-439). Since biomarkers of responsiveness to immunotherapies are so rare and gender is not sufficiently considered in immunology studies, we would prefer to leave this result in the manuscript. We have added further experiments to mechanistically pursue the result referred to in Figure 1g, showing an enhanced response in the liver compared to blood. We have provided new data showing increased ACAT efficacy in a high cholesterol media, of relevance to its increased effects on T-cells from the liver, the central hub of cholesterol metabolism (new SuppFig1t and SuppFig2d; results p7 line 133-137, p8 line 161-165, discussion p19 line 436-439).

- Fig1F – what peptide pool is these data referring to? Legend indicates that data shows HBV OLP +/- inhibitor compared to unstimulated control. Do authors mean normalised to unstimulated control? And compared with and without inhibitor, as the graph seems to suggest? This is confusing (please also refer to my comment number 1)

The data in Fig 1f refers to HBV-derived OLP pools spanning HBcAg, HBsAg and polymerase; we have now added this information to the results (p6 line 119) as well as legend. The data is presented as fold increase of IFN γ production after stimulation with HBV-derived OLP or HBV-derived OLP+ACAT inhibition, now clarified in the legend. This format of data presentation is in line with previous studies evaluating the effect of immunotherapy (e.g. Day et al, Nature 2006) but we also complement it with the format in SuppFig 1n where we show the actual % response for each donor with or without ACAT inhibition (after background subtraction as explained above).

- Fig4D shows the results of only one experiment (representative of two) with a really modest increase in IFN γ expression. I do not think the data supports the conclusion. More repeats and/or another readout for IFN γ production would be required.

To address this important point raised by reviewer 1+2, we have further extended the data on HBV TCR-gene-modified T-cells (new Figure 4d and SuppFig.4l-o, results p14 line 301-306). We have now confirmed that ACAT inhibition increased the number IFN γ -producing cells and the amount of IFN γ produced per cell by intracellular cytokine staining (ICS) in four independent experiments (Fig4d, SuppFig4l). The increase IFN γ was confirmed by Luminex supernatant analysis (SuppFig4m). Further analysis revealed that ACAT inhibition also enhanced the production of other immune

mediators with potential antitumour role, as assessed by ICS (TNF, MIP1 β ; SuppFig 4n,o) and by Luminex (eg. TNF, MIP1 β , GranzymeB, Flt-3L, CD40L; SuppFig 4m). We have moved the more preliminary data on HA-1-specific TCR-gene-modified T-cells to the supplementary figure (SuppFig 4p).

The scale of these changes is subtle because of the use of very low peptide concentrations and hepatoma cell lines with poor presentation efficiency to mimic the limitations on T-cell target recognition in the liver and tumours, where small increases in T-cell functional avidity at limiting concentrations are likely to be biologically critical. - Definitions of HBc (core) and HBs are missing (they are probably very familiar to hepatitis B experts, but not to the general audience). A table with the peptides used might also be informative.

We have included these definitions in the text (p6 line 119). The peptides used are all OLP spanning the whole of the relevant proteins rather than selected HLA-restricted epitopes; we have now added the number of peptides in each pool in the methods (p22 line 479-485).

Reviewer 2:

....Although the mechanism part to some extent is not of high novelty, the ex vivo studies with HBV/HCC patient T cells make this work highly interesting and valuable for developing new therapeutics against HBV and HBV-related HCC. I only have a few minor comments.

We thank this reviewer for their enthusiastic appreciation of the translational value of our study using precious ex vivo HBV/HCC patient T-cells.

1. In addition to use measure raft-associated cholesterol enriched by CTB, the authors shall use the ALOD4 probe to measure whether free cholesterol level of the plasma membrane could be influenced by ACAT inhibition.

We were unable to obtain the probe mentioned so have instead used Filipin complex from *Streptomyces filipinensis* to detect free membrane cholesterol. We observed a significant increase of free membrane cholesterol after treatment with ACAT inhibition in high cholesterol media (new SuppFig 2d, results p8 line 161-165, discussion p19 lines 436-439). Further supporting the observed higher efficacy of ACAT inhibition on T-cells isolated from the high cholesterol environment of the liver compared to their circulating counterpart T-cell responses (now shown for paired CD4⁺T-cells, new SuppFig 1s, as well as CD8⁺T-cell responses to HBV, Fig 1g; results p7 line 130-137).

2. For the HCC TIL analysis, why in some patients ACAT inhibition induced contrary effect under different TAA stimulation?

As previously described, TAA expression and TAA-specific responses in patients with HCC are heterogenous (e.g. Flecken et al, *Hepatology* 2014). Different mechanisms/stages of exhaustion, different exposure to the tumour microenvironment and different affinity and avidity may explain the observed differences of response to ACAT inhibition of T-cells with different specificity in the same patient. As discussed in the response to point 4 from Reviewer 1, *in vitro* studies will tend to underestimate therapeutic responses achieved *in vivo*, but even *in vivo* responses are typically restricted to subsets of cohorts in chronic viral infections and tumours due to the multi-faceted nature of T-cell constraints in these diseases. Other immunotherapeutic strategies such as PD-1 blockade have shown that *in vitro* responses in some donors remain unchanged or even decrease (perhaps due to activation-induced cell death) as also seen with ACAT inhibition. However, as we show in Fig. 3e, the proportion of responders is actually higher for ACAT inhibition with or without PD-1 blockade than for PD-1 blockade alone, whilst ACAT rescue of TILs is highly consistent, suggesting this will be an advance on the existing limited response to PD-1 blockade in HCC clinical trials.

3. Figure 4D, 4E and S4D lack statistical analysis.

To address this important point raised by reviewer 1+2, we have further extended the data on HBV TCR-gene-modified T-cells (new Figure 4d and SuppFig.4l-o, results p14 line 301-306). We have now confirmed that ACAT inhibition increased the number IFN γ -producing cells and the amount of IFN γ produced per cell by intracellular cytokine staining (ICS) in four independent experiments with statistical analysis (Fig4d, Supp Fig4l). This increase was confirmed by Luminex supernatant analysis (SuppFig4m). Further analysis revealed that ACAT inhibition also enhanced the production of other immune mediators with potential antitumour role, as assessed by ICS (TNF, MIP1b; SuppFig 4n,o) and by Luminex (eg. TNF, MIP1 β , GranzymeB, Flt-3L, CD40L; SuppFig 4m). We have moved the more preliminary data on HA-1-specific TCR-gene-modified T-cells to the supplementary figure (SuppFig 4p).

Reviewer 3:

.....authors proposed that ACAT inhibition provided a paradigm of a metabolic checkpoint that is able to constrain tumours and viruses but rescue exhausted T cells, rendering it an attractive therapeutic target for the functional cure of HBV and HBV-related HCC. However, the evidence is inadequate to support this conclusion,

and some of the data required more experiments to verify since the difference between experiment groups was not significant. Statistical analysis should be shown in the figure legend of figures and supplementary figures.

The statistical analysis is described at the end of each figure legend and in the methods section. In experiments with low n numbers (<50) we performed non-parametric tests. In experiments with n>50, a D'Agostino & Pearson normality test was applied; since normality was not achieved, non-parametric tests were again applied. The statistical test was chosen accordingly (Wilcoxon matched-pairs signed rank test, Mann-Whitney test, Friedman test with Dunn's multiple comparisons test, Kruskal-Wallis test with Dunn's multiple comparisons test, Fisher's exact test, Spearman correlation) with significant differences marked on all figures. All tests were performed as two-tailed tests, and for all tests, significance levels were defined as not significant (ns) $P \geq 0.05$; * $P < 0.05$; ** $P < 0.01$; *** $P < 0.001$; **** $P < 0.0001$. We have clarified the statistical methodology in the relevant Methods section (p28, line 638-647).

1. Inhibition of SOAT1 (also known as ACAT) markedly reduced the size of tumours that had high levels of SOAT1 expression. In this study, authors described ACAT a dual viral and T-cell metabolic checkpoint. More evidence is required to clear explain the mechanisms of ACAT behaviour in immune cells in addition to the tumor cells.

ACAT1 is known to be expressed by T-cells, explaining their susceptibility to ACAT inhibitors and some mechanisms by which they respond to ACAT inhibitors have been defined in murine cells by Yang et al (Nature 2016). We add to this by showing the conserved activity of ACAT inhibitors in human tissue samples from liver and HCC and by defining further relevant metabolic pathways including increased lipid microdomains, reduced lipid droplets and altered TCR-independent bioenergetics (OXPHOS, glycolysis). We agree that further detailed analysis of indirect metabolic changes such as alteration in transcription factors (e.g. SREBP) would be interesting for future studies as mentioned in the discussion (p19 line 417-423).

2. How about the ACAT metabolism pathway in intrahepatic CD8+T cells during chronic HBV infection? And what are the main factors derived from HBV or HCC that are responsible for the altered ACAT? These are important for explore the mechanisms of ACAT in CD8+T cells.

We thank the reviewer for prompting us to add this important angle on differential ACAT transcription in the disease settings studied. The capacity of ACAT inhibition to directly inhibit tumour cell proliferation and migration in HCC was linked to higher ACAT1 in tumour cells of responders (Jiang et al, Nature 2019). We have now investigated whether T-cells in the settings we have examined have differential expression of ACAT that might contribute to any preferential responsiveness to pharmacological inhibition of this pathway. We analysed published single cell transcriptomic data from human blood, liver and HCC (Zheng et al Cell 2017); ACAT1 (SOAT1) transcripts were detectable in comparable proportions of intrahepatic compared to peripheral CD4⁺ and CD8⁺T-cells and in marginally higher proportions of TIL than in the periphery (new SuppFig 1k,l and SuppFig 4b,c, results p6 line 112-116 and p12 line 267-270), whereas ACAT2 was barely detectable, as expected.

Single-cell RNA-seq liver data has shown that ACAT is also detectable in healthy human hepatocytes and different intrahepatic immune cells including macrophages and T-cells (MacParland, Nat Comm 2018). We have now analysed available microarray data for bulk liver samples and found that ACAT1 transcripts are increased in CHB compared to healthy and further increased in the setting of HBV-related liver inflammation (Zhou, Liver Int 2017; Wang Sci Rep 2017; Liu, J Inf Dis 2018, added to new SuppFig5c-e, results p16 line 355-360). These data provide further possible support for further studies to investigate whether there is preferential antiviral and immunomodulatory activity of ACAT inhibitors in the HBV infected liver *in vivo*.

As an additional explanation for the enhanced response of intrahepatic and intratumoural T-cells, we provide new data showing that T-cells cultured in higher cholesterol concentrations (reflective of the liver and tumours) become more responsive to ACAT inhibition (new SuppFig 1t, results p7 lines 133-137, discussion p19 line 436-438). Moreover, we have shown that low CD28 expression, a feature of T-cells in CHB, tumours and other states of T-cell exhaustion, associates with low lipid microdomain staining and an enhanced capacity of ACAT inhibition to boost T-cell responses (Fig3 c,d and SuppFig 3b).

3. To demonstrate ACAT as an attractive therapeutic target for the functional cure of HBV and HBV-related HCC, experiments *in vivo* need to be under investigation. For example, HBV-carrier mouse model, acute HBV infection mouse model, and HBV-related HCC mouse model may be used to observe the effects of ACAT inhibition *in vivo*.

We agree that *in vivo* HBV and HCC models would be useful for further study of the therapeutic potential of ACAT inhibition. While this is beyond the scope of this submitted manuscript, we hope that our manuscript will encourage further *in vivo* studies; we have now referred to the need for these on p16 line 353-354. We have also changed the title of the manuscript to "Targeting human Acyl-CoA:cholesterol acyltransferase as a dual viral and T-cell metabolic checkpoint" to highlight that this manuscript does not include the study of animal models.

4. IFN- γ secretion is only one of the functional molecules of CD8⁺T cells. More work need to be performed to explore the mechanisms of ACAT inhibition in CD8⁺T cells.

We had already included proliferation (CFSE dilution), TNF production and degranulation (CD107a mobilisation) as additional readouts for circulating HBV-specific T-cells, although we found IFN γ to be the most sensitive marker for non-cytolytic antiviral function, as supported by multiple previous HBV studies in humans. In response to this reviewer's request, we have now added TNF and CD107a data for intrahepatic HBV-specific T-cell responses (new SuppFig1 p,q; results p6 line 124-126) and intratumoral TAA-specific responses (new SuppFig 4g,h; results p13 line 282-283).

We have also added further analysis of TCR-gene-modified T-cells, revealing that ACAT inhibition also enhanced the production of other immune mediators with potential antitumour role, as assessed by intracellular cytokine staining (TNF, MIP1 β ; SuppFig4 n,o; results p14 line 303-306) and by supernatant analysis by Luminex (eg. TNF, MIP1 β , GranzymeB, Flt-3L, CD40L; Supp Fig4m; results p14 line 303-306).

5. HCC related antigens were described. How to select these antigens? It was wonder whether they are highly expressed in HCC tumor cells from the patients?

We agree with the reviewer that an analysis of TAA expression in each patient's tumour and stimulation of TIL with the matching TAA would be ideal; however, this was not possible in our study due to the limited amount of tumour tissue available from each patient, which was all required for isolation of lymphocytes for analysis of low-frequency T-cell responses. We therefore chose well-described TAA that are highly expressed in HCC tumour tissue (Breous J Hepatol 2011), and have been shown to elicit TAA-specific CD8⁺T-cell responses (Flecken Hepatology 2014) that are associated with increased survival in HCC (Flecken, Hepatology 2014). Additionally, the cancer-testis antigens NY-ESO-1 and MAGE-A1 are shared TAA expressed by various cancer types, including melanoma, and therefore make our findings potentially transferable to other tumour types.

6. As shown in figures, the frequency of IFN- γ +CD8⁺T cells were still in a low level in the groups of ACAT inhibition. What about the absolute numbers of IFN- γ +CD8⁺T cells? The effects of these increased CD8⁺T cells during the liver injury should be investigated further.

The magnitude of increase seen with these very low frequency HBV and HCC-specific T-cells in our study is subtle but is in the order of other *in vitro* studies of immunotherapeutic strategies that have progressed through to the clinic, as discussed in point 4 to reviewer 1. As a proxy measure for changes in absolute numbers we have now added a figure showing that ACAT inhibition also increased the number of IFN γ -producing CD8⁺T-cells when analysed as frequency of total CD45⁺ cells isolated from HBV-infected liver (SuppFig 1n; results p6 line 121-122).

The reviewer raises the important point that all immunotherapeutic approaches boosting antiviral T-cells in chronic HBV carry a risk of liver injury since T-cells can also trigger bystander liver injury. Encouragingly, ACAT inhibition preferentially enhanced the non-cytolytic (IFN γ) rather than cytolytic (CD107a) antiviral potential of intrahepatic HBV-specific CD8⁺ IHL (new SuppFig 1q, results p6 line 12-126), which is the type of response described in animal models and human studies to favour HBV clearance with minimal liver damage. Additionally, in clinical studies on hyperlipidaemia and/or (cardio-)vascular disease, no severe hepatotoxicity was reported for the ACAT inhibitor Avasimibe (*Insull, Atherosclerosis 2001; Raal, Atherosclerosis 2003; Tardif, Circulation 2004; Hiatt, Vasc Med 2004*); however, patients with pre-existing hepatic dysfunction were excluded from these studies and any clinical study exploring the use of ACAT inhibitors in patients with CHB will need to carefully monitor hepatotoxicity and immune-mediated liver injury.

7. In supplemental figure 4, the frequency of intratumoral CD3⁺T cells was very low. How about their absolute numbers? Further, intratumoral CD8⁺T cells deserved investigating for their function with ACAT inhibition or not.

As above, we have now included the proxy measure for absolute numbers of TAA-specific T-cells expressed as a percentage of total CD45⁺ cells isolated from HCC and from liver tissue surrounding the tumour (new SuppFig4 d,k; results p13 line 276-277, 286-2867). The frequency of TAA-specific CD8⁺T-cells is very low, as expected; other CD8⁺T-cells within HCC TILs are expected to be directed against neoantigens not tested here. The reviewer refers to the frequency of intratumoral CD3⁺T-cells, which we had not included previously; we have now added supplementary figures showing that the frequency of global CD3⁺T-cells and CD8⁺T-cells within these tumours is actually quite high although very variable (ranging from 19-80% and 3-31% respectively) and is not affected by ACAT inhibition (new SuppFig 4f, results p13 line 279-280).

8. In this study, authors determined the ACAT metabolism in CD8⁺T cells. During the development of HCC, other immune cells such as NK cells, NKT cells, $\gamma\delta$ T cells, which were predominant cell populations in the liver, played critical anti-tumor activities. Does ACAT inhibition affect their functions?

We thank the reviewer for the important suggestion of studying other immune cells contributing to the immune microenvironment. We treated PBMC from patients with CHB with ACAT inhibitors and stimulated them with IL-12/IL-18 (n=14) or a pan- $\gamma\delta$ -antibody (n=11) to study the effect of ACAT inhibition on NK cell and $\gamma\delta$ T-cell function, respectively. We did not detect any significant changes of cytokine production (IFN γ , TNF) or degranulation (CD107a mobilization) by NK cells or $\gamma\delta$ T-cells (global and $v\delta 2$). However, we feel that further experiments and extensive optimization (drug concentration, treatment duration, method of stimulation) would be necessary to reliably determine if there is any effect of ACAT inhibition on these other immune cells. We would therefore prefer not to include these data in the revised manuscript.

However, to extend the scope of this manuscript beyond CD8⁺T-cells, we did also investigate the effect of ACAT inhibition on CD4⁺T-cells. ACAT inhibition led to a significantly increased function of HBV- and TAA-specific CD4⁺T-cells isolated from the liver and tumour, respectively, and a trend to increased HBV-specific IFN γ production in the blood. We have included these data in the manuscript (SuppFig 1f,r,s and SuppFig 4i, results p5 line 98-99; p6 line 126-128; p7 line 130-132; p13 line 283-284).

We hope that with the changes detailed above, the manuscript will now be suitable for publication in Nature Communications. All authors concur with these additions, which are highlighted in red throughout the manuscript.

Yours sincerely,

Mala Maini PhD FRCP FMedSci
Professor and Honorary Consultant, Viral Immunology
Wellcome Trust Senior Investigator
Division of Infection and Immunity UCL
Rayne Building Room 420, 5 University St
London WC1E 6JF
Tel +44 (0)20 3108 2170 <https://www.ucl.ac.uk/maini-group>

REVIEWER COMMENTS

Reviewer #1 (Remarks to the Author):

Dear Authors and Editors

I have now gone through all the changes made by the authors regarding the comments that I suggested and I am pleased to see that all of them have been addressed in the best possible manner. Hence, in my opinion, the work is in much better shape now and I am recommending this for publication in your journal.

Espe

Reviewer #2 (Remarks to the Author):

My previous concerns have been successfully addressed in the revised manuscript.

Chenqi Xu
Shanghai Institute of Biochemistry and Cell Biology
Chinese Academy of Sciences

Reviewer #3 (Remarks to the Author):

Targeting human Acyl-CoA:cholesterol acyltransferase as a dual viral and T-cell metabolic checkpoint is an interesting issue, however, the conclusions in this paper should be confirmed by experiments in vivo. At present, all the data of ACAT inhibition were obtained by experiments in vitro with cell culture. ACAT therapeutic effects on actual anti-HBV activity and anti-tumor activity are not well demonstrated in this paper, although some significant changes in the frequency, number, functional molecules were observed in T cells by ACAT inhibition. Additionally, it is important to show the main factors derived from HBV-infected hepatocytes/HCC tumor cell/T cells that are responsible for the altered ACAT, which will provide necessary clues for targeting ACAT.

Our response to the Reviewers is as follows:

Reviewer 1:

Thank you for recommending the work for publication.

Reviewer 2:

Glad to hear your previous concerns have been successfully addressed.

Reviewer 3:

As already agreed with the Editor after the first submission, *in vivo* animal work is beyond the scope of this study. The reviewer is mistaken in referring to anti-tumour effects needing better demonstration as our study did not examine direct anti-carcinogenic effects of ACAT inhibition but referred to previous studies already demonstrating this for HCC (*Jiang et al, Nature 2019*). The direct effects on HBV particle genesis we show are novel and do need conformation in future *in vivo* studies, as we have discussed in the manuscript. In this revision we have further highlighted the *in vitro* nature of the antiviral effects demonstrated in the abstract and 2 sentence summary.

In our revised submission we included a new body of work at this Reviewer's request, examining differential transcription of ACAT in different T cells and HBV vs non-HBV-infected livers and the influence of local cholesterol concentrations; we do not think that further exploration of these factors is essential to progress the application of ACAT inhibitors in these clinical settings. Moreover, such work constitutes a different study quite distinct to our successful demonstration of ACAT activity in human tissue T cells directed against HBV and HCC.